


# Compound coastal flood risk in a semi-arid urbanized region: The implications of copula choice, sampling, and infrastructure

Joseph T.D. Lucey[1] and Timu W. Gallien[1]

[1]Department of Civil and Environmental Engineering, University of California, Los Angeles, CA 90095, USA

**Correspondence:** Timu Gallien (tgallien@seas.ucla.edu)

**Abstract.** Sea level rise will increase the frequency and severity of coastal flooding events. Compound coastal flooding is characterized by multiple flooding pathways (i.e., high offshore water levels, streamflow, energetic waves, precipitation) acting concurrently. This study explores the joint flood risks caused by the co-occurrence of high marine water levels and precipitation in a highly urbanized semi-arid, tidally dominated region. A novel structural function developed from the multivariate analysis

is proposed to consider the implications of flood control infrastructure in compound coastal flood risk assessments. Univariate statistics are analyzed for individual sites and events. Conditional, and joint probabilities are developed using a range of copulas and sampling methods. The Independent, and Cubic copulas produced poor results while the Fischer-Kock, and Roch-Alegre generally produced robust results across a range of sampling methods. The impacts of sampling are considered using annual maximum, annual coinciding, wet season monthly coinciding, and wet season monthly maximum sampling. Although, annual

maximum sampling is commonly recommended for characterizing compound events, this work suggests annual maximum sampling does not produce "worst-case" event pairs and substantially underestimates marine water levels for extreme events. Wet season coinciding water level and precipitation pairs benefit from a dramatic increase in available data, improved goodness of fit statistics, and provide a range of physically realistic pairs. Wet season coinciding sampling may provide a more accurate compound flooding risk characterization for long return periods in semi-arid regions.

# 1 Introduction

Coastal flooding is a significant human hazard (Leonard et al., 2014; Wahl et al., 2015) and is considered a primary health hazard by the U.S. Global Change Research Program (Bell et al., 2016). Coastal migration and utilization continues to increase (Nicholls et al., 2007; Nicholls, 2011). Over 600 million people populate coastal zones (Merkens et al., 2016). Climate change-induced sea level rise will substantially increase flood risk (Church et al., 2013; Horton et al., 2014), and negatively impact

coastal populations (Bell et al., 2016). Even relatively modest sea level rise will significantly increase flood frequencies through the US (e.g., Tebaldi et al., 2012; Taherkhani et al., 2020). Southern California is particularly vulnerable to sea level rise. Small changes in sea level (∼5 cm) double the odds of the 50-year flooding event (Taherkhani et al., 2020) and the 100-year event is expected to become annual by 2050 (Tebaldi et al., 2012). Regional research has explored flood risks caused by sea level rise and coastal forcing (e.g., Heberger et al., 2011; Hanson et al., 2011; Gallien et al., 2015). However, accurately





characterizing future, non-stationary coastal vulnerability requires considering the joint and potentially nonlinear impacts of compound (marine and hydrologic) events (Gallien et al., 2018).

Compound coastal flooding considers the combined impacts of marine and hydrologic forcing. Typical events, such as precipitation or high water levels, occurring simultaneously may combine to generate extreme events (Seneviratne et al., 2012). In urban coastal settings multiple flooding pathways (i.e., high marine water levels, wave runup and overtopping, large fluvial

flows, and pluvial flooding from precipitation) interact with infrastructure (e.g., sea walls, human-made dunes, and the storm system) potentially exacerbating hazards. Notably, these events may produce vastly different flooding outcomes for an event of a given return period. Traditionally, literature has focused on river discharge or storm surge dominated compound events (Table 1).

From a flood risk perspective there are multiple methods to characterize potential events. A univariate approach is often used

where a single variable (e.g., water level) is considered. For example, in coastal flooding events FEMA suggests a univariate approach where marginal statistics are developed, independently modelled, and the more severe result is adopted (FEMA, 2011, 2016c). Ironically, this can lead to underestimating flood risk because of the interplay between two flood pathways (i.e., a high tail water forces fluvial flooding upstream). Conditional probabilities represent an alternative where the compound flood risk can be evaluated given available information on a primary variable (e.g., water level) to determine the exceedance

probability of a secondary variable (e.g., precipitation) (Shiau, 2003; Karmakar and Simonovic, 2009; Zhang and Singh, 2012; Li et al., 2013; Mitková and Halmová, 2014; Serinaldi, 2015, 2016; Anandalekshmi et al., 2019). A third method uses copulas to analyze the dependence of multiple flood drivers and develop joint statistics.

Numerous studies have used a copula based approach to study floods manifested by various combinations of variables (Table 1). Compound flood risks can be described and quantified from previous copula studies (Salvadori, 2004; Salvadori

and De Michele, 2004, 2007; Salvadori et al., 2011, 2013, 2016). Multivariate inland and coastal hydrology analysis have primarily focused on a small group of copulas: Archimedean (Clayton, Frank, and Gumbel), Student t, and Gaussian copulas. Alternative copulas may more accurately characterize urban coastal flooding (Jane et al., 2020). Specifically, hazard scenarios provide various perspectives on critical multivariate events (Salvadori et al., 2016). Current studies are often limited to select hazard scenarios (Table 1 in Salvadori et al. (2016)).

Data sampling methods in multivariate studies influence distribution fitting. Two primary sampling methods exist: peaks over threshold (Jarušková and Hanek, 2006) and block maxima (Engeland et al., 2004). Events selected using peaks over threshold sampling are above a predetermined threshold defining an "extreme" event. Block maxima sampling uses various block sizes (yearly, seasonal, semiannual, etc.) to separate and select the maximum event per block. Engeland et al. (2004) reports a significantly different 1,000-year streamflow when using 12-month block sampling (160 m$^3$s$^{-1}$) compared to using a

threshold of 50 m$^3$s$^{-1}$ (120 m$^3$s$^{-1}$). Many studies utilize block maxima sampling with a yearly block size, i.e. the annual maximum sampling method (Baratti et al., 2012; Bezak et al., 2014; Wahl et al., 2015). This method is specifically recommended by FEMA (2016c) for evaluating coastal hazards. Alternatively, studies identify extremes in the primary variable using annual maximum sampling and select a secondary variable which co-occurs with the primary variable (Lian et al., 2013; Xu et al., 2014; Tu et al., 2018), creating a "coinciding" type sampling. Multivariate applications relying upon annual maximum



sampling generate a "worse case" scenario which may produce unrealistic variable combinations. Coinciding sampling draws from physically realistic event pairs. Although, multiple studies explore sampling effects on fitted distribution parameters and univariate return periods (Engeland et al., 2004; Jarušková and Hanek, 2006; Peng et al., 2019; Juma et al., 2020), sampling effects for multivariate coastal flooding events is unknown.

    Coastal flooding studies primarily focus on locations defined by storm surge dominated oceanographic conditions with
warm, humid (Wahl et al., 2012; Lian et al., 2013; Xu et al., 2014; Masina et al., 2015; Wahl et al., 2015; Mazas and Hamm, 2017; Paprotny et al., 2018; Tu et al., 2018; Bevacqua et al., 2019; Didier et al., 2019; Xu et al., 2019; Yang et al., 2020), and monsoonal (Jane et al., 2020) climatic conditions. In contrast, along the southern California coast typical tidal variability is 1.7 to 2.2 m (Flick, 2016) and storm surge rarely exceeds ∼20 cm (Flick, 1998). Notably, during the wet season (October to March), when precipitation typically occurs, spring tide ranges are relatively large (∼ 2.6 m). Critically, few studies consider
areas where coastal flooding events are dominated by large tides and either precipitation or wave events (Masina et al., 2015; Mazas and Hamm, 2017; Didier et al., 2019; Jane et al., 2020). This study explores univariate and multivariate flooding events in a semi-arid, tidally dominated, highly urbanized region. Here, the dependency between observed water levels and precipitation, impacts of sampling methods and distribution fitting, and the most likely flood values are explored.

## 2   Site description & data

This study considers observed water level and precipitation influences for coastal compound events at Santa Monica (SM), Sunset Beach (S), and LA Jolla (SD) areas in Los Angeles, Huntington Beach, and San Diego, California (Fig. 1); three semi-arid, tidally dominated sites in the US. All are low-lying estuarine or bay-backed highly urbanized beach communities requiring extensive coastal management to mitigate flooding events. For example, sea walls and artificial berms in Sunset Beach protect infrastructure from high embayment water levels, wave runup, and overtopping along the open coast. The storm drain network
is managed to prevent back flooding during high tides. Notably, Gallien et al. (2014) suggested when tide valves are closed, the storm drain network cannot remove pluvial flooding caused by alternative flooding pathways (e.g., precipitation or wave overtopping). Pacific Coast Highway (PCH) is heavily utilized and is a primary transportation corridor along the southern California coastline. All locations are densely urbanized and highly impacted by flooding.

    Observed water levels from the Los Angeles (Station ID: 9410660), La Jolla (Station ID: 9410230), and Santa Monica (Sta-
tion ID: 9410840) tide gauges are available on NOAA's Tides and Currents at various observation frequencies (NOAA, Accessed 2019c). Verified hourly water levels (m NAVD88) were downloaded with varying records. Hourly, instead of high/low, data were used to benefit from the extended record (additional 31-years for Los Angeles and La Jolla, and 6-years for Santa Monica). It is worth noting, that in current compound flooding literature, the terms tide and storm surge are often interchanged. Recent efforts have been made to standardize language where tide represents only the astronomical changes in water levels and
storm surge specifically excludes astronomical variability and consists only of the inverse barometric effects along with wind and wave setup (Gregory et al., 2019). In this study, the term observed water level (OWL) is adopted. OWL is the water level measured at the NOAA tide gauges which includes all tidal, storm, and climatic effects.





**Table 1.** List of multivariate studies which utilized copulas to study the associated variables

| Variable Pairs | References |
| --- | --- |
| Waves and water level | Masina et al. (2015); Mazas and Hamm (2017); Didier et al. (2019) |
| Waves and storm duration | De Michele et al. (2007); Salvadori et al. (2014, 2015) |
| Waves and storm surge | Wahl et al. (2012); Paprotny et al. (2018) |
| River discharge and water level | White (2007); Bray and McCuen (2014); Sadegh et al. (2018) |
| River discharge and storm surge | Paprotny et al. (2018) |
| River discharge and volume | Yue (2001a, b); Shiau (2003); Favre et al. (2004); De Michele et al. (2005); Poulin et al. (2007); Li et al. (2013); Salvadori et al. (2013); Requena et al. (2013); Aghakouchak (2014) |
| River discharge, rainfall, and water level | Bray and McCuen (2014); Jeong et al. (2014) |
| Multiple river discharges | Salvadori and De Michele (2010) |
| Rainfall and tide | Lian et al. (2013)[*], Xu et al. (2014)[*], Tu et al. (2018)[*], Xu et al. (2019)[*], Yang et al. (2020)[*] |
| Rainfall and water levels | Jane et al. (2020) |
| Rainfall and storm surge | Wahl et al. (2015); Paprotny et al. (2018); Bevacqua et al. (2019) |
| Rainfall intensity and depth | Yue (2000a, b, 2002); De Michele and Salvadori (2003) |
| Rainfall and groundwater | Anandalekshmi et al. (2019) |
| Rainfall and runoff | Zhang and Singh (2012); Hao and Singh (2020) |
| Rainfall and river discharge | Zhong et al. (2020) |
| Rainfall and temperature | Zhang et al. (2017) |
| Rainfall and duration | Salvadori and De Michele (2007) |
| Combinations of rainfall intensity, depth, and duration | Zheng et al. (2014) |
| Combinations of river discharge, volume, and duration | Karmakar and Simonovic (2009); Gräler et al. (2013); Mitková and Halmová (2014) |

*Note these studies use the term tide measurement but actually represent observed water level measurements. Please refer Section 2 for clarification.

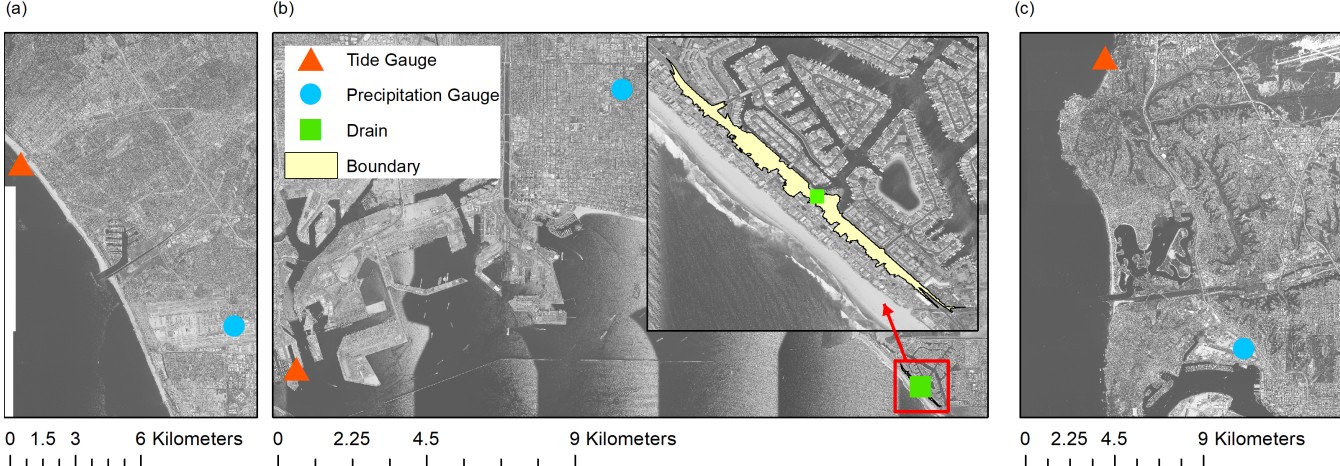

**Figure 1.** Map displaying (a) Santa Monica, (b) Sunset, and (c) San Diego sites along with locations of tide gauges (triangle) and precipitation stations (circle). The road drain (square) and boundary (yellow) at Sunset ($\sim 2\,\mathrm{km}^2$) is for the Structural scenario. Aerial imagery from NOAA 2020.

The U.S. Hourly Precipitation Data dataset provided by NOAA's National Centers for Environmental Information (NOAA, Accessed 2019b) at the Signal Hill (COOP:048230), Los Angeles International Airport (COOP:045114), and San Diego International Airport (COOP:047740) stations is used as the precipitation inputs. Observations do not contain trace amounts ($< 0.25\,\mathrm{mm}$) and is provided as cumulative precipitation ($\mathrm{mm}$) per event. Precipitation measurements were converted to a $\mathrm{mm/hr}$ rate to match the hourly OWL measurements. The final precipitation input is a 24-hour cumulative precipitation record made from the hourly observations. All data was transformed to UTC for analysis.

Compound flood probabilities are determined with combinations of sampling methods: Annual Maximum (AM), Annual Coinciding (AC), Wet Season Monthly Maximum (WMM), and Wet Season Monthly Coinciding (WMC). A summary of each sites' associated gauges, observation windows, and number of pairs is provided in Table 2. Southern California's wet season is defined between October to March and provides a majority of the total annual rainfall (Cayan and Roads, 1984; Conil and Hall, 2006). It is likely for extreme compound events to occur during this period. Maximum sampling pairs the single largest precipitation and OWL observations within a year or month, a "worse case scenario" approach. Coinciding sampling pairs the single largest precipitation observation within a year or month to the largest OWL observation within its 24-hour accumulation period, providing more realistic pairs compared to maximum sampling.

Distributions are fit with existing precipitation observations greater than zero consistent with previous studies (Swift Jr and Schreuder, 1981; Hanson and Vogel, 2008). Months with no OWL measurements were also excluded. In the case of coinciding sampling, pairs that had three or more OWL measurements missing within the 24-hour window were manually reviewed and removed if the tidal peak was clearly missing. Specifically for WMM sampling, months with more than half its observations missing were also reviewed and removed if the tidal peak was missing. The resulting data pairs are shown in Fig. 2.





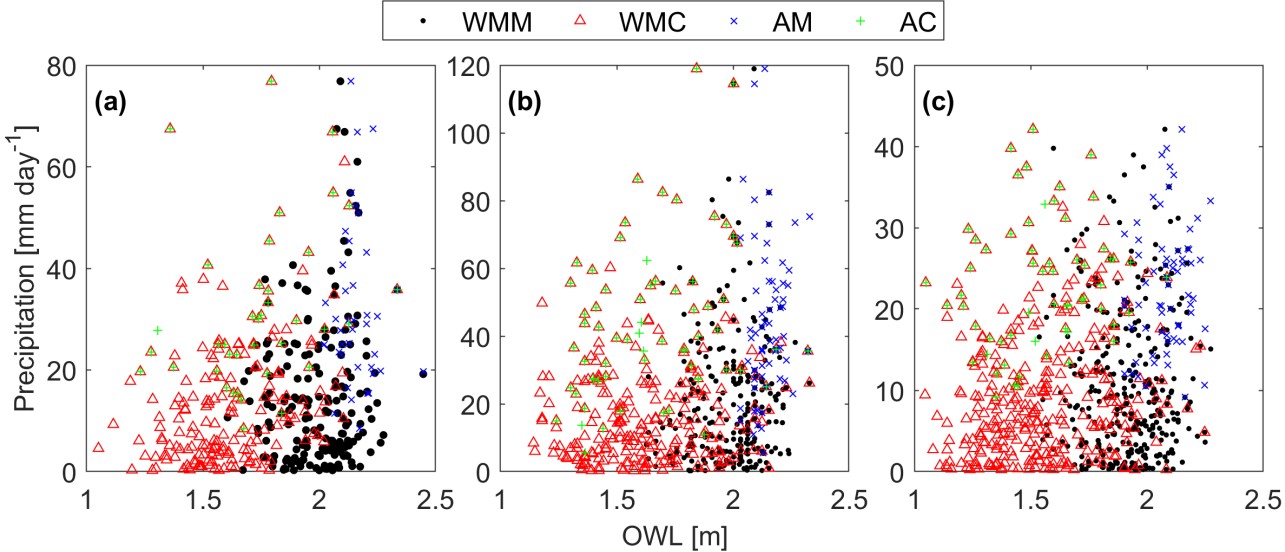

**Figure 2.** Data pairs for each sampling method (AM = cross; AC = plus; WMM = dot; WMC = triangle) at (a) SM, (b) S, and (c) SD

**Table 2.** Water level and precipitation observations at Santa Monica (SM), Sunset (S), and San Diego (SD)

| Site | Tide Gauge | Precip. Gauge | Observation Window | AM Pairs | AC Pairs | WMM Pairs | WMC Pairs |
|------|-----------|---------------|--------------------|----------|----------|-----------|-----------|
| SM | 9410840 | 045114 | 22 November 1973 to 19 December 2013 | 40 | 38 | 193 | 191 |
| S | 9410660 | 048230 | 01 July 1948 to 01 December 2012 | 63 | 63 | 257 | 258 |
| SD | 9410230 | 047740 | 01 July 1948 to 19 December 2013 | 65 | 60 | 328 | 329 |

## 3 Methods

### 3.1 Univariate, bivariate, & conditonal distributions

Potential flooding events are determined with three different probability definitions: univariate, conditional, and bivariate. As-
115 suming $X$ and $Y$ are random variables, $x$ and $y$ are observations of these variables, and $F_X$ and $F_Y$ represent the variables' re-
spective cumulative distribution functions (CDF). Formulations for univariate ($F_X(x)$, $F_Y(y)$) and bivariate joint ($F_{XY}(x,y)$)
CDFs follow DeGroot and Schervish (2014) (Eq. (1) and (2)). Conditionals ($F_{X|Y \geq y}(x|Y \geq y)$, $F_{X|Y \leq y}(x|Y \leq y)$, and
$F_{X|Y=y}(x|Y=y)$) are developed from Shiau (2003) (Eq. (3)) and Serinaldi (2015) (Eq. (4) and (5)). Conditionals 1 (C1),
2 (C2), and 3 (C3) represent Eq. (3), (4), and (5) going forward. Univariate statistics are developed using the appropriate
continuous random variable distribution while conditional and bivariate CDFs are determined using copulas.

Copulas are functions that associate random variables' univariate CDFs to their joint CDF (e.g., $F_X$ and $F_Y$ to $F_{X,Y}(x,y)$)
according to Sklar's theorem (Sklar, 1959; Salvadori, 2004). There is no requirement for the univariate distributions to be





the same. This is particularly advantageous since the optimal univariate distributions may be used for each variable. Bivariate probabilities for different hazard scenarios, which represent various multivariate events, and conditional probabilities can be

calculated using fitted copula functions.

$$F_X(x) = Pr(X \leq x) \tag{1}$$

$$F_{X,Y}(x,y) = Pr(X \leq x \ and \ Y \leq y) \tag{2}$$

$$F_{X|Y \geq y}(x|Y \geq y) = Pr(X > x|Y \geq y) = \frac{F_X(x) - F_{XY}(xy)}{1 - F_Y(y)} \tag{3}$$

$$F_{X|Y \leq y}(x|Y \leq y) = Pr(X > x|Y \leq y) = 1 - \frac{F_{X,Y}(x,y)}{F_Y(y)} \tag{4}$$

$$F_{X|Y=y}(x|Y=y) = Pr(X > x|Y=y) = 1 - \frac{\partial F_{XY}(xy)}{\partial y} \tag{5}$$

### 3.2    Hazard scenarios

Notation and definitions from Salvadori et al. (2016), unless otherwise stated, is used to define the upper set ($S$) and scenario types. Salvadori et al. (2016) and Serinaldi (2015) present figures of each scenario's probability space. Further discussion of hazard scenarios and copulas assume a bivariate situation.

### 3.2.1    "OR"

"OR" scenario events have one or both random variables exceed a specified threshold. That is, what is the probability of a water level or precipitation event exceeding a given value? Standard univariate CDFs make up the associated copula.

$$\alpha_x^\vee = \mathbf{P}(\mathbf{X} \in S_x^\vee) = 1 - \mathbf{C}(\mathbf{F}_1(x_1), ..., \mathbf{F}_d(x_d)) \tag{6}$$

### 3.2.2    "AND"

"AND" scenario events have both random varibles exceed a specified threshold. In this case the fundamental question is "what is the probability of a particular water level and precipitation rate exceeding specified values?". The survival copula ($\hat{\mathbf{C}}(u,v)$) is comprised of univariate survival CDFs ($\bar{\mathbf{F}}(x) = 1 - \mathbf{F}(x)$) and the provided equation can be found in Serinaldi (2015) and Salvadori and De Michele (2004).

$$\alpha_x^\wedge = \mathbf{P}(\mathbf{X} \in S_x^\wedge) = \hat{\mathbf{C}}(\bar{\mathbf{F}}_1(x_1), ..., \bar{\mathbf{F}}_d(x_d)) \tag{7}$$





$$\hat{\mathbf{C}}(u,v) = 1 - u - v + C(u,v) \tag{8}$$

### 3.2.3 "Kendall"

"Kendall" (K) scenario highlights an infinite set of OR events that separate "safe" and "dangerous" statistical regions. In the OR scenario, events along an isoline ($t$) share a common probability, but define separate regions. Events along a Kendall $t$ represent the same super critical region (Serinaldi, 2015) and provide a "safety lower bound" (Salvadori et al., 2011). Essentially the Kendall considers the minimum OR events of concern. $\mathbf{K}$(t) is estimated by a method outlined in Salvadori et al. (2011).

$$\mathbf{K}(t) = \mathbf{P}(\mathbf{F}(X_1,...,X_d) \leq t) = \mathbf{P}(\mathbf{C}(F_1(X_1),...,F_d(X_d)) \leq t) \tag{9}$$

$$\alpha_t^{\mathbf{K}} = \mathbf{P}(\mathbf{X} \in S_t^{\mathbf{K}}) = 1 - \mathbf{K}(t) \tag{10}$$

### 3.2.4 "Survival Kendall"

"Survival Kendall" (SK) scenario highlights an infinite set of AND events which also separate safe and dangerous statistical spaces. AND events along a $t$ also share a common probability, but define separate regions. Events along an SK $t$ represent the same super critical region, but provide an "(upper) bounded safe region" (Salvadori et al., 2013). The Survival Kendall specifically considers the largest AND events of concern and is estimated by the method outlined in Salvadori et al. (2013).

$$\hat{\mathbf{K}}(t) = \mathbf{P}(\overline{\mathbf{F}}(X_1,...,X_d) \leq t) = \mathbf{P}(\hat{\mathbf{C}}(\bar{F}_1(X_1),...,\bar{F}_d(X_d)) \leq t) \tag{11}$$

$$\alpha_t^{\check{\mathbf{K}}} = \mathbf{P}(\mathbf{X} \in S_t^{\check{\mathbf{K}}}) = 1 - \check{\mathbf{K}}(t) = \hat{\mathbf{K}}(t) \tag{12}$$

### 3.2.5 "Structural"

"Structural" scenario considers the probability of an output from a structural function, $\Psi(\mathbf{X})$, exceeding a design load or capacity ($z$). In this work, the structural failure functions focuses on the question "what is the probability of a water level forcing tide valve closure and subsequent flooding during a precipitation event?".

$$\alpha_z^{\Psi} = \mathbf{P}(\mathbf{X} \in S_z^{\Psi}) = \mathbf{P}(\Psi(\mathbf{X}) > z) \tag{13}$$



### 3.3 MvCAT

The Multivariate Copula Analysis Toolbox (MvCAT) developed by Sadegh et al. (2017) is a publicly available MATLAB toolbox that fits 25 different copula functions to user data of two random variables. Copula parameters are optimized through a local optimization or with Markov Chain Monte Carlo methods (details in Sadegh et al. (2017)). The MvCAT framework is expanded to determine all scenarios and conditionals in this study. While all copulas have functional CDFs, the Cuadras-Auge, Raftery, Shih-Louis, Linear-Spearman, Fischer-Hinzmann, Husler-Reiss, Cube, and Marshal-Olkin copulas do not have a PDF function. A PDF is required to determine the most likely value along an isoline, therefore removing those copulas from the study. Copulas must also be computationally simple to derive or integrate to calculate Conditional 3. Gaussian and Student t copulas' partial derivatives cannot be explicitly calculated, and estimates induce unrealistic errors (i.e., produce negative probabilities). Seventeen different copulas remain after eliminating those discussed above.

### 3.4 Return periods

Hydrologic events are commonly cast in the context of return periods (e.g., De Michele et al., 2005, 2007; FEMA, 2011; Wahl et al., 2012; USACE, 2013; Salvadori et al., 2014; Wahl et al., 2015; Salvadori et al., 2015). Return periods ($T$) provide an easily interpreted metric describing the severity of an event and is the inverse of an event's probability of exceedance presented as $F$ in Eq. (14) (Tu et al., 2018). In Eq. (15), $N$ is the data's time window, $n$ is the number of considered events within $N$, and $N_e$ is the average number of events per unit of time (monthly, yearly, etc.). Therefore, $N_e = 1$ when considering singular events within a year (Tu et al., 2018).

$$T = 1/(N_e * F) \tag{14}$$

$$N_e = n/N \tag{15}$$

### 3.5 Goodness of fit metrics

Multiple goodness of fit metrics and correlations serve to quantify the quality of distribution fits and dependencies between variables. Marginal and copula fits are selected by Bayesian Information Criterion (BIC; Eq. (19)) and maximum likelihood (ML; Eq. (16)) values, respectively. Likelihood ($\mathcal{L} \in [0, \infty)$) measures how well a distribution's estimated parameters fit the sample data with larger values suggesting a better fit. Log-likelihood ($\ell \in (-\infty, \infty)$) is the log transformation of Eq. (16) used to calculate BIC. BIC ($BIC \in (-\infty, \infty)$) is similar to the likelihood, but penalizes for the number of estimated parameters ($D$) and the data's sample size ($n$). Smaller BIC values represent a better fit. Equations and definitions can be found in Sadegh et al. (2017). Correlation measurements include Pearson's linear correlation, Kendall's tau, and Spearman's rho coefficients.

$$\mathcal{L}(\boldsymbol{\theta}|\tilde{\mathbf{Y}}) = \prod_{i=1}^{n} \frac{1}{\sqrt{2\pi\tilde{\sigma}^2}} exp\{-\frac{1}{2}\tilde{\sigma}^2[\tilde{y}_i - y_i(\boldsymbol{\theta})]^2\} \tag{16}$$





$$\tilde{\sigma}^2 = \frac{\sum_{i=1}^{n}[\tilde{y}_i - y_i(\boldsymbol{\theta})]^2}{n} \tag{17}$$

$$\ell(\boldsymbol{\theta}|\tilde{\mathbf{Y}}) = -\frac{n}{2}\ln(2\pi) - \frac{n}{2}\ln\tilde{\sigma}^2 - \frac{1}{2}\tilde{\sigma}^2\sum_{i=1}^{n}[\tilde{y}_i - y_i(\boldsymbol{\theta})]^2 \tag{18}$$

$$BIC = Dln(n) - 2\ell \tag{19}$$

## 4 Results

Univariate, conditional, and bivariate probabilities were developed using four sampling methods (AC, AM, WMC, WMM) and seventeen different copulas. Figure 3 presents marginal distribution results for all sites and sampling. Two marginal distributions do not pass the chi square test at the standard 0.05 level of significance (San Diego AM OWL and Santa Monica WMM OWL).

These distributions pass at reduced significance levels of 0.01. Additionally, Santa Monica's AM data is slightly negatively correlated ($> -0.06$). Copula and sampling effects differ significantly at low (i.e., low return period) and high (i.e., severe return period) probabilities of non-exceedance. In the case of annual sampling, non-exceedance (exceedance) probabilities are 0.9 (0.1) and 0.99 (0.01) for the 10- and 100-year events, respectively. In wet season sampling, return period exceedance probabilities vary depending on sampling type and location due to the average number of event observations per year ($N_e$ from

Eq. (15)). For example, San Diego WMC sampling has 329 observations within the 65 year record (i.e., $N_e = 5.06$). Therefore, the exceedance probabilities ($F$ in Eq. (14)) associated to a 10- and 100-year event are 0.0198 and 0.0020 (non-exceedance probabilities at 0.9802 and 0.9980), respectively. Table 3 presents wet season (WMM and WMC) exceedance probabilities for all sites.

**Table 3.** Santa Monica, Sunset, and San Diego exceedance probabilities at the 10- and 100-year return periods for WMM and WMC samplings.

|  | Santa Monica | | Sunset | | San Diego | |
|---|---|---|---|---|---|---|
|  | 10-year | 100-year | 10-year | 100-year | 10-year | 100-year |
| WMM | 0.0207 | 0.0021 | 0.0245 | 0.0025 | 0.0198 | 0.0020 |
| WMC | 0.0209 | 0.0021 | 0.0244 | 0.0024 | 0.0198 | 0.0020 |

### 4.1 Marginals

The selected marginal distributions (Fig. 3, Table 4) were tested and/or suggested fits in previous studies. Rainfall has been widely fit with an Exponential distribution (refer to Table 2 in Salvadori and De Michele (2007)), but more recently been fit


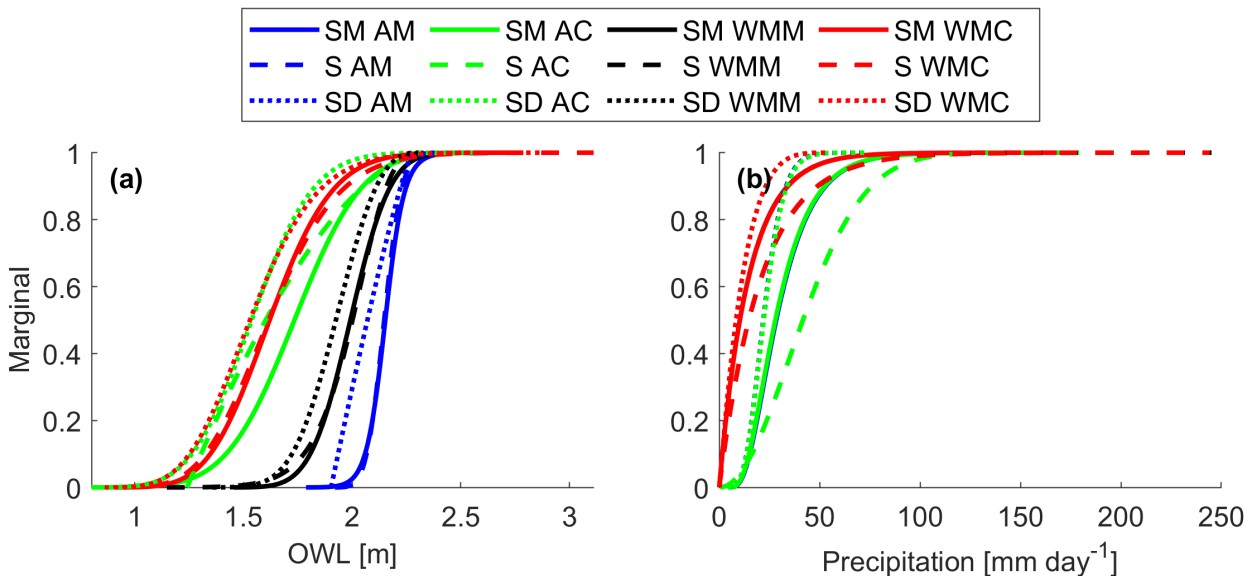

**Figure 3.** (a) OWL and (b) precipitation marginals for Santa Monica (solid lines), Sunset (dashed lines), and San Diego (dotted lines) using AM (blue), AC (green), WMM (black), and WMC (red) samplings.

using a variety of distributions including Gamma (Husak et al., 2007), Rayleigh (Pakoksung and Takagi, 2017; Esberto, 2018), Generalized Pareto or Birnbaum-Saunders (Ayantobo et al., 2021). In the case of annual precipitation sampling (coinciding or maximum) Santa Monica was well described by a Birnbaum-Sanders, Rayleigh best described Sunset data, and a Gamma was

the best fit for San Diego data. Similarly, wet season precipitation data (maximum or coinciding) was best described by the Exponential distribution for Santa Monica and Sunset while the Generalized Pareto best represented San Diego.

Historically, water levels have been described using a number of distributions including Normal (Hawkes et al., 2002), Generalized Pareto (Mazas and Hamm, 2017), Log Logistic and Nakagami (Sadegh et al., 2018), Birnbaum-Saunders (Sadegh et al., 2018; Didier et al., 2019; Jane et al., 2020), along with Gamma, Weibull, and Inverse Gaussian (Jane et al., 2020).

Observed water levels did not exhibit site specific patterns and were described by a range of distributions (Table 4).

### 4.2 Copulas

The quality of a copula's fit is determined by its maximum likelihood (ML) values. Figure 4 presents ML values for the 17 tested copulas at each site (columns) per sampling method (rows). Fits produced similar ML statistics for a given site and sampling method, however, a few copulas produced particularly poor or unusual results. For example, the Independent and Cubic were

consistently ranked as the lowest fit. Additionally, a number of copulas generally perform well (Fig. 4). For example, the Fischer-Kock (Fisc.), Tawn, and Roch-Alegre (Roch.) were amongst the better fitting copulas, ranking within the top five ML for nearly all datasets. The Tawn has previously been used in a OWL and precipitation analysis (Jane et al., 2020) and in this





**Table 4.** Best fitting univariate distributions for each location and sampling method.

| Dataset | Variable | Santa Monica | Sunset | San Diego |
|---------|----------|--------------|--------|-----------|
| AM | OWL | L | BS | GP |
|    | Precip | BS | R | G |
| AC | OWL | N | GP | NA |
|    | Precip | BS | R | G |
| WMM | OWL | N | W | GEV |
|     | Precip | E | E | GP |
| WMC | OWL | G | IG | IG |
|     | Precip | E | E | GP |

BS - Birnbaum-Saunders; GP - Generalized Pareto; E - Exponential

R - Rayleigh; N - Normal; L - Log logistic; G - Gamma

W - Weibull; IG - Inverse Gaussian; NA - Nakagami

case, is frequently the highest ranked. All other copula results were within the envelopes described by the five copulas. Unless otherwise noted, these focused copulas will be used to consider the impacts of copula choice and sampling.

San Diego WMC conditional CDFs display the impacts of individual copulas (Fig. 5). The Independent and Cubic copulas consistently suggest lower OWL (Fig. 5a, e) and precipitation values (Fig. 5b, f) while, in this example, the Tawn suggests substantially higher OWL and precipitation values (dashed line, Fig. 5a, b, e, f). C1's 100-year pair in Table 6 displays an example of the Tawn's conservative nature. The Roch-Alegre and Fischer-Kock provide highly consistent results for both precipitation and water level (black and green lines, Fig. 5a, b, e, f). Copula choice has nearly no effect on the Conditional

2 scenario (Fig. 5c, d). Most probable OWL and precipitation values in Tables 5 and 6 further display the aforementioned behaviors. These conditional patterns generally persist at all locations with an additional note that the Tawn, Roch-Alegre, and Fischer-Kock copulas typically provide similar results.

Figures 6 and 7 show the 10- and 100-year return periods, respectively, for the five focused copulas using WMC sampling at SD. In all the hazard scenarios, the Independent and Cubic copulas suggest most probable events dominated by either large

precipitation or OWL (Fig. 6 and 7). For example, AND 100-year OWL is unrealistically small compared to the other OWL values provided by alternative copulas (Table 6). Clearly the Tawn presents conservative results suggesting higher OWL and precipitation pairs in the AND scenario (Fig. 6a and 7a), while the Roch-Alegre and Fischer-Kock copulas present intermediate OWL and precipitation values. Similarly for the Survival Kendall scenario, the Tawn displays more conservative OWL and precipitation pairs while the Roch-Alegre and Fischer-Kock suggest the moderate OWL and precipitation levels (Fig. 6c and

7c). The OR and Kendall scenarios suggest quite similar isolines with the most probable precipitation varying between copulas (Fig. 6b, d, and 7b, d). Tables 5 and 6 further display the bivariate patters. Again, these bivariate patterns generally persist at all locations with an additional note that the Tawn, Roch-Alegre, and Fischer-Kock, copulas often generate similar values.

**Figure 4.** ML Values per fitted copula for SM (left column), S (middle column), and SD (right column) (a), (b), (c) AM, (d), (e), (f) AC, (g), (h), (i) WMM, and (j), (k), (l) WMC

**Figure 5.** SD WMC OWL (left column) and precipitation (right column) (a), (c) C1, (c), (d) C2, and (e), (f) C3 CDFs using the Independent (blue), Cubic (red), Roch-Alegre (green), Fischer-Kock (black), and Tawn (black dashed) copulas. OWL/ Precipitation conditionals are conditioned on the occurrence of a 25-year precipitation/ OWL event.

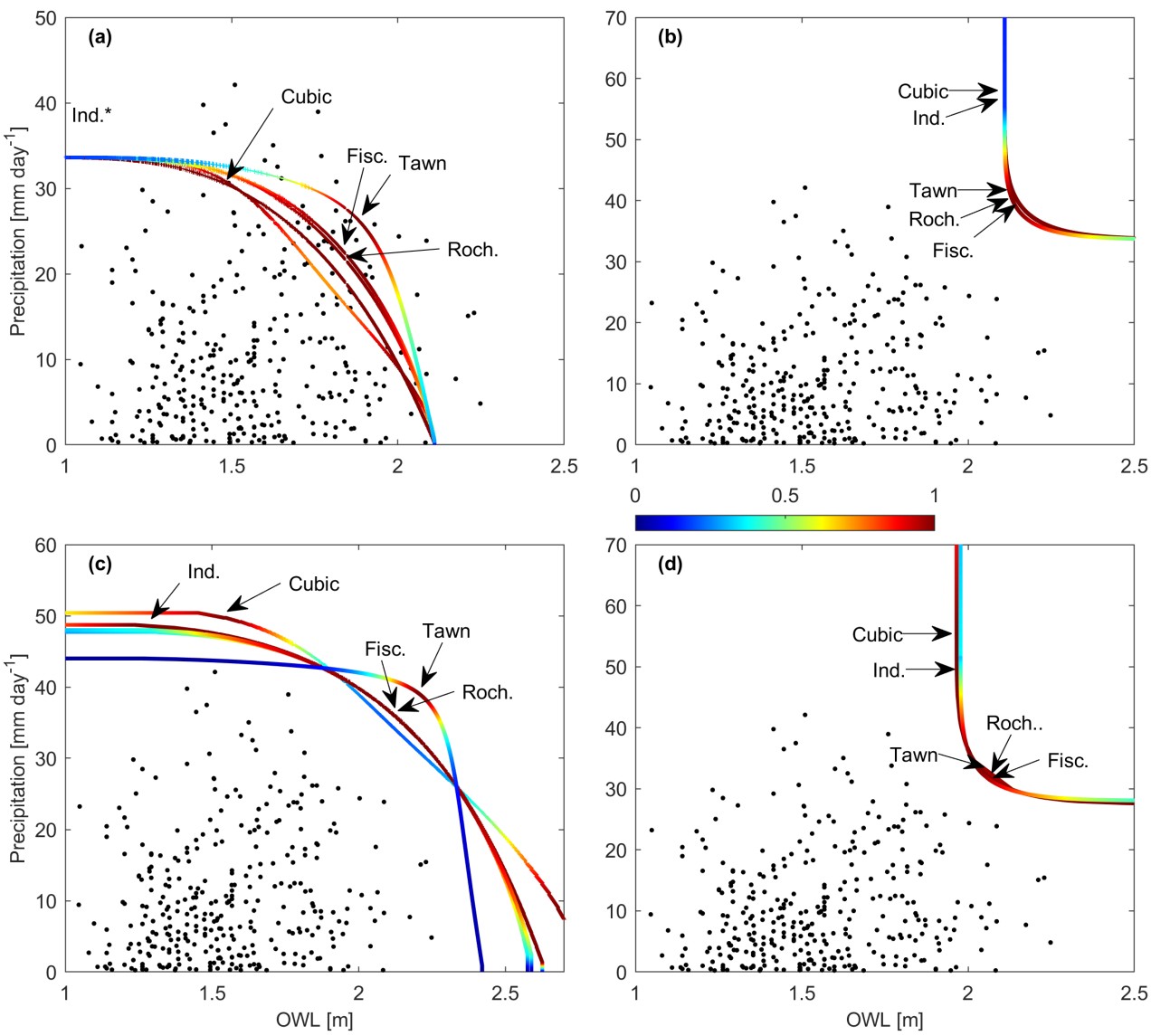

**Figure 6.** SD WMC (a) AND, (b) OR, (c) SK, and (d) K hazard scenarios with the Independent, Cubic, Roch-Alegre, Fischer-Kock, and Tawn 10-year isolines. Copula labels point to the mostly likely value on their respective isolines. Most likely values outside the axis have a * with their respective copula label.

**Figure 7.** SD WMC (a) AND, (b) OR, (c) SK, and (d) K hazard scenarios with the Independent, Cubic, Roch-Alegre, Fischer-Kock, and Tawn 100-year isolines. Copula labels point to the mostly likely value on their respective isolines. Most likely values outside the axis have a * with their respective copula label.





**Table 5.** SD 10-year marginal, conditional, and bivariate OWL (m) and precipitation ($\mathrm{mmday}^{-1}$) values using WMC sampling. Conditionals are conditioned on a 25-year event occurring.

|  | Ind | | Cubic | | Roch. | | Fisc | | Tawn | |
| --- | --- | --- | --- | --- | --- | --- | --- | --- | --- | --- |
|  | OWL | Precip | OWL | Precip | OWL | Precip | OWL | Precip | OWL | Precip |
| M | 2.11 | 33.64 | 2.11 | 33.64 | 2.11 | 33.64 | 2.11 | 33.64 | 2.11 | 33.64 |
| C1 | 2.11 | 33.64 | 1.94 | 26.34 | 2.18 | 36.27 | 2.16 | 35.57 | 2.59 | 48.57 |
| C2 | 2.11 | 33.64 | 2.11 | 33.68 | 2.11 | 33.61 | 2.11 | 33.62 | 2.09 | 33.07 |
| C3 | 2.11 | 33.64 | 1.94 | 26.59 | 2.18 | 36.24 | 2.16 | 35.56 | 2.40 | 44.73 |
| AND | 0.74 | 33.64 | 1.51 | 30.07 | 1.85 | 22.15 | 1.84 | 21.80 | 1.89 | 26.09 |
| OR | 2.11 | 55.32 | 2.11 | 55.32 | 2.17 | 38.29 | 2.20 | 37.06 | 2.14 | 39.10 |
| K | 1.97 | 48.34 | 1.97 | 55.32 | 2.06 | 31.79 | 2.06 | 31.80 | 2.02 | 34.43 |
| SK | 1.23 | 48.72 | 1.54 | 49.71 | 2.15 | 35.11 | 2.15 | 35.06 | 2.21 | 38.87 |

**Table 6.** SD 100-year marginal, conditional, and bivariate OWL (m) and precipitation ($\mathrm{mmday}^{-1}$) values using WMC sampling. Conditionals are conditioned on a 25-year event occurring.

|  | Ind | | Cubic | | Roch. | | Fisc. | | Tawn | |
| --- | --- | --- | --- | --- | --- | --- | --- | --- | --- | --- |
|  | OWL | Precip | OWL | Precip | OWL | Precip | OWL | Precip | OWL | Precip |
| M | 2.40 | 43.34 | 2.40 | 43.34 | 2.40 | 43.34 | 2.40 | 43.34 | 2.40 | 43.34 |
| C1 | 2.40 | 43.34 | 2.18 | 36.43 | 2.46 | 44.95 | 2.44 | 44.53 | 2.85 | 52.72 |
| C2 | 2.40 | 43.34 | 2.40 | 43.37 | 2.40 | 43.32 | 2.40 | 43.34 | 2.36 | 42.07 |
| C3 | 2.40 | 43.34 | 2.20 | 36.88 | 2.46 | 44.92 | 2.44 | 44.51 | 2.62 | 49.38 |
| AND | 0.74 | 43.38 | 1.53 | 41.02 | 2.03 | 30.45 | 2.02 | 30.06 | 2.18 | 38.04 |
| OR | 2.39 | 55.32 | 2.39 | 55.32 | 2.48 | 45.39 | 2.47 | 45.53 | 2.43 | 46.54 |
| K | 2.13 | 55.32 | 2.15 | 42.63 | 2.22 | 37.67 | 2.17 | 40.29 | 2.17 | 39.81 |
| SK | 1.73 | 51.49 | 2.79 | 13.98 | 2.33 | 41.53 | 2.32 | 41.26 | 2.47 | 46.35 |

## 4.3 Sampling

San Diego conditional CDFs using the Fischer-Kock copula clearly present sampling effects (i.e. maximum versus coinciding and annual versus wet season months). Coinciding samplings exhibit similar OWL CDFs (green and red lines, Fig. 8a, c, e), where as wet season samplings exhibit similar precipitation CDFs (red and black lines Fig. 8b, d, f). OWL values are generally larger for maximum sampling at lower non-exceedance probabilities (i.e lower return periods) (Fig. 8a, c, e blue and black lines, Table 7). However, the extended tail from WMC sampling produces larger OWL at higher return periods (red line Fig. 8a, c, e; Table 8). Annual coinciding sampling displays significantly lower OWL values at low (Table 7) and high (Table 8) return periods. Only minimal differences between annual and wet season precipitation exist at the 10- and 100-year return





periods (maximum difference of 1.79 and 3.08 $\mathrm{mmday^{-1}}$ respectively; Tables 7 and 8). Annual (wet season) precipitation CDFs appear similar as OWL measurements are chosen subsequent to precipitation observations (Fig. 8b, d, f).

Figures 9 and 10 present the 10- and 100-year return periods using the Fischer-Kock copula for all samplings (AC, AM, WMC, WMM). For the AND and K scenarios, AM sampling results in the largest OWL compared to the other sampling methods (Fig. 9a, d and 10a, d, Tables 7 and 8). Additionally for the AND and K scenarios (Fig. 9a, d and 10a, d), maximum samplings (AM and WMM) provide more conservative OWLs compared to WMC OWL values (Tables 7 and 8). When comparing maximum samplings (AM and WMM) to WMC sampling in the OR and SK scenarios, maximum samplings generally provide larger OWL values at lower return periods (Fig. 9b, c; Table 7), but smaller or similar OWL at larger return periods (Fig. 10b, c; Table 8). AC sampling generally results in the smallest OWL levels at all hazard scenarios. These behaviors persist across all locations, suggesting the maximum type sampling may not accurately reflect OWL at extreme return periods for univariate and conditional situations.

Table 7 and 8 shows the most probable 10- and 100-year marginal and compound event values. AM OWL exceed AC OWL across all probability types and sites, which is expected given the (nonphysical) paring of the two largest individual OWL and precipitation events without regard to co-occurrence. For example, in the 10-year return period AM OWLs are at least 30 $\mathrm{cm}$ higher than AC (Table 7). In the 100-year return period AM exceeds AC by at least 17 $\mathrm{cm}$ (Table 8). Precipitation is generally consistent across all conditionals, scenarios and sampling types with only minor variations observed. The AND scenario is the exception to this where substantially lower precipitation values are seen across all sampling and and scenarios. AM and WMM sampling generally produced similar OWL results at both the 10- and 100-year return periods with maximum difference of 6 $\mathrm{cm}$ across all conditionals and copulas.

**Table 7.** SD 10-year marginal, conditional, and bivariate OWL (m) and precipitation ($\mathrm{mmday^{-1}}$) values using the Fischer-Kock. Conditionals are conditioned on a 25-year event occurring.

|  | AM | | AC | | WMM | | WMC | |
|  | OWL | Precip | OWL | Precip | OWL | Precip | OWL | Precip |
|---|---|---|---|---|---|---|---|---|
| M | 2.22 | 33.38 | 1.83 | 33.76 | 2.20 | 33.61 | 2.11 | 33.64 |
| C1 | 2.23 | 33.78 | 1.86 | 35.10 | 2.20 | 34.23 | 2.16 | 35.57 |
| C2 | 2.22 | 33.37 | 1.83 | 33.70 | 2.20 | 33.60 | 2.11 | 33.62 |
| C3 | 2.23 | 33.77 | 1.86 | 35.07 | 2.20 | 34.23 | 2.16 | 35.56 |
| AND | 2.14 | 26.07 | 1.66 | 26.77 | 2.09 | 20.89 | 1.84 | 21.80 |
| OR | 2.25 | 37.33 | 1.91 | 37.49 | 2.22 | 38.03 | 2.20 | 37.06 |
| K | 2.24 | 36.03 | 1.89 | 36.44 | 2.18 | 32.14 | 2.06 | 31.80 |
| SK | 2.25 | 38.22 | 1.94 | 38.78 | 2.21 | 35.01 | 2.15 | 35.06 |

**Figure 8.** SD OWL (left column) and precipitation (right column) (a), (b) C1, (c), (d) C2, and (e), (f) C3 CDFs for AM (blue), AC (green), WMM (black), and WMC (red) samplings using the Fischer-Kock copula. OWL/ Precipitation conditionals are conditioned on the occurrence of a 25-year precipitation/ OWL event.

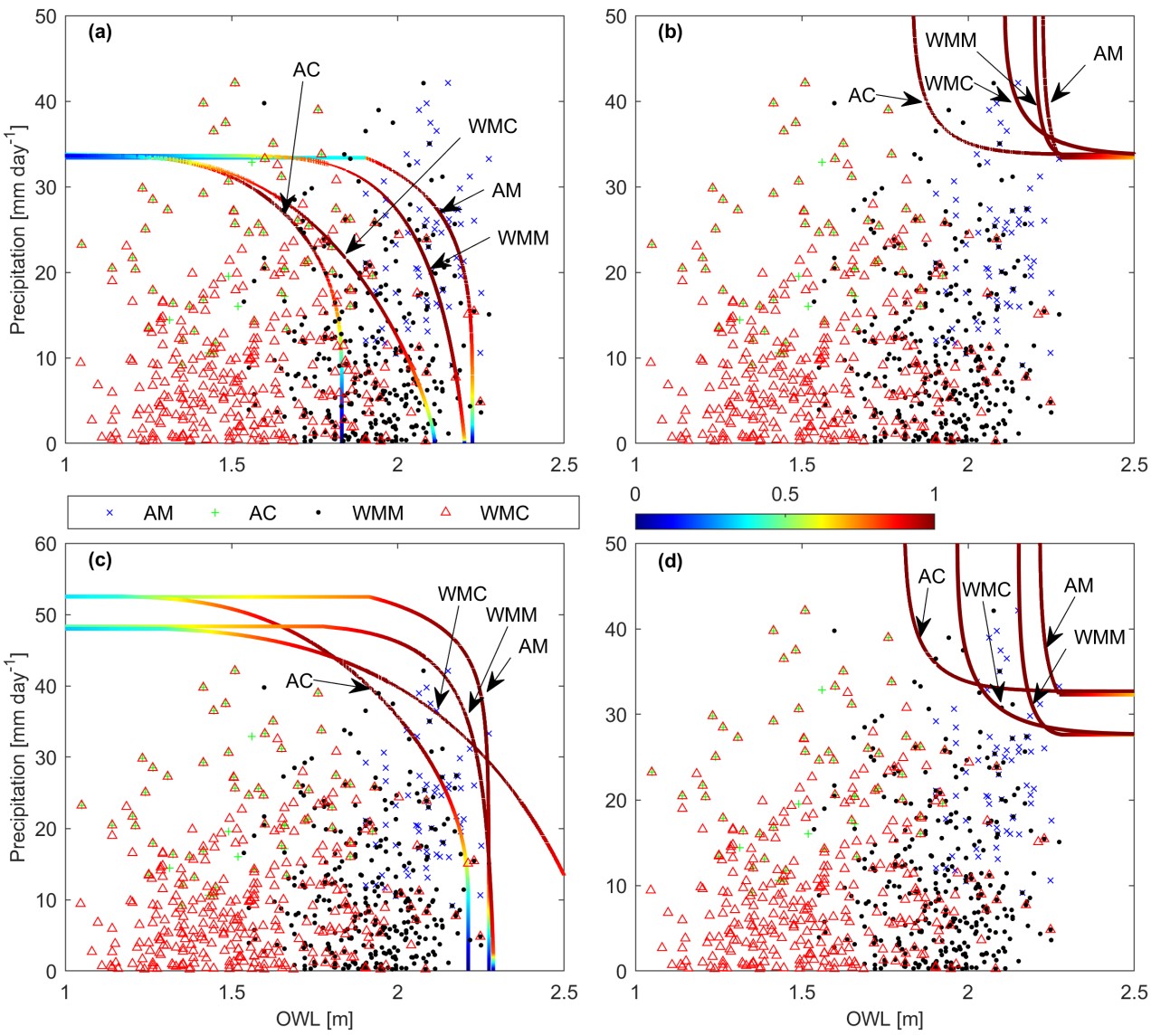

**Figure 9.** SD (a) AND, (b) OR, (c) SK, and (d) K hazard scenarios for AM (cross), AC (plus), WMM (dot), WMC (triangle) data and 10-year isolines using the Fischer-Kock copula. Sampling labels point to the mostly likely value on their respective isolines.

## 4.4 Structural failure

A unique structural scenario is presented to consider flood severity along the Pacific Coast Highway (PCH) in Sunset Beach. PCH road elevation ranges from 1.7-2.4 m NAVD88 (Fig. 11), below typical spring tide (∼2.13 m) and more extreme (∼2.3 m) water levels (NOAA, Accessed 2019a), requiring tide valves along PCH for flood prevention. Tide valve closures prevent


**Figure 10.** SD (a) AND, (b) OR, (c) SK, and (d) K hazard scenarios for AM (cross), AC (plus), WMM (dot), WMC (triangle) data and 100-year isolines using the Fischer-Kock copula. Sampling labels point to the mostly likely value on their respective isolines.

back-flooding from high bay water levels coming up through subsurface storm drains that (normally) discharge to the bay. Additionally, closed tide valves also enable precipitation pooling since water cannot be drained to the bay. Severe pooling may result in a critical highway closure, which can further damage property and inhibit emergency service operations.

Areal precipitation flooding extent and depth can be estimated for water levels exceeding tide valve closure elevation. A water level equal or greater than 1.68 m NAVD88 forces valve closures and frames the structural failure as a Conditional 1



**Table 8.** SD 100-year marginal, conditional, and bivariate OWL (m) and precipitation ($\mathrm{mmday}^{-1}$) values using the Fischer-Kock. Conditionals are conditioned on a 25-year event occurring.

| | AM | | AC | | WMM | | WMC | |
|---|---|---|---|---|---|---|---|---|
| | OWL | Precip | OWL | Precip | OWL | Precip | OWL | Precip |
| M | 2.27 | 45.07 | 2.08 | 45.65 | 2.26 | 43.32 | 2.40 | 43.34 |
| C1 | 2.27 | 45.40 | 2.10 | 46.78 | 2.26 | 43.70 | 2.44 | 44.53 |
| C2 | 2.27 | 45.06 | 2.08 | 45.59 | 2.26 | 43.32 | 2.40 | 43.34 |
| C3 | 2.27 | 45.40 | 2.10 | 46.75 | 2.26 | 43.70 | 2.44 | 44.51 |
| AND | 2.23 | 33.59 | 1.85 | 34.43 | 2.16 | 29.27 | 2.02 | 30.06 |
| OR | 2.27 | 58.78 | 2.09 | 60.14 | 2.27 | 45.52 | 2.47 | 45.53 |
| K | 2.26 | 43.02 | 2.02 | 42.45 | 2.22 | 39.32 | 2.17 | 40.29 |
| SK | 2.27 | 45.05 | 2.08 | 45.63 | 2.25 | 40.76 | 2.32 | 41.26 |

type event. The local watershed is convex and drains an area of 94,897 m². Water pools in the low elevation areas along PCH (Fig. 11). When pluvial water levels exceed the sea wall elevation, water overflows the sea wall and exits to the harbor. The maximum pool storage is 11,342 m³. The percent of flooding is then calculated with Eq. (20) as the structural function.

Structural scenario precipitation and percent flooding ($\Psi$) values utilizing the Roch-Alegre, Fischer-Kock, and Tawn copulas are shown in Table 9. Rows and columns separate the utilized sampling methods and return periods of interest, respectively. Fig. 12 shows $\Psi$ as a function of precipitation for the Roch-Alegre, Fischer-Kock, and Tawn copulas along with the 5- (square), 10- (circle), and 100-year (diamond) return periods. All copulas display similar values across sampling methods with the exception of the Tawn, which presents more conservative values using WMC sampling (Fig. 12, Table 9). For example, the 10-year WMC for the Tawn (Fig. 12c) is over 80 percent flooding compared to approximately 70 percent with the other copulas. Again, AC sampling severely underestimates precipitation and flooding.

$$\% \; flooding = \frac{precipitation \times area}{volume} \times 100 \tag{20}$$

## 5   Discussion

Previous multivariate studies typically use a small, popular group of copulas (e.g., Clayton, Frank, Gumbel, Student t, and Gaussian). Gaussian and Student t copulas were excluded due to their lack of a computationally simple derivative or integral while the Clayton, Frank, and Gumbel generally provided average fits (Fig. 4). Tawn, Fischer-Kock, and Roch-Alegre copulas generally present similar values with the Tawn occasionally presenting more conservative pairs (Fig. 7). Cubic and Independent copulas consistently present event pairs predominantly over- or underestimating OWL and precipitation (Fig. 7). The poor performance of the Independent copula further emphasizes the importance in accounting for dependencies between variables in multivariate studies (Raynal-Villasenor and Salas, 1987; Yue and Rasmussen, 2002; De Michele et al., 2005). Well fit copulas




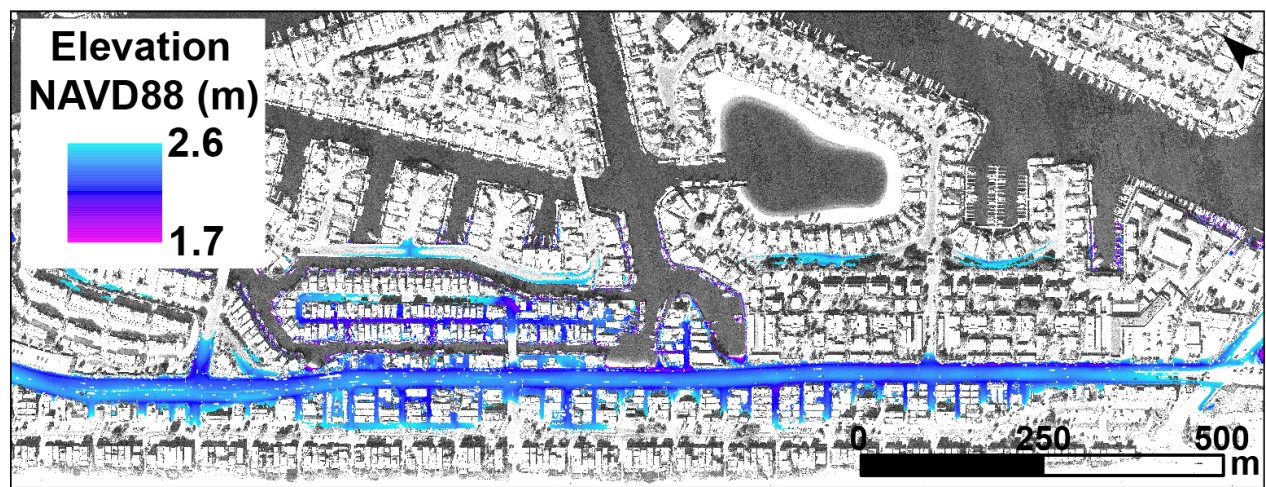

**Figure 11.** Elevations within the PCH boundary ranging from low (purple) to high (blue).Background imagery from NOAA 2020.

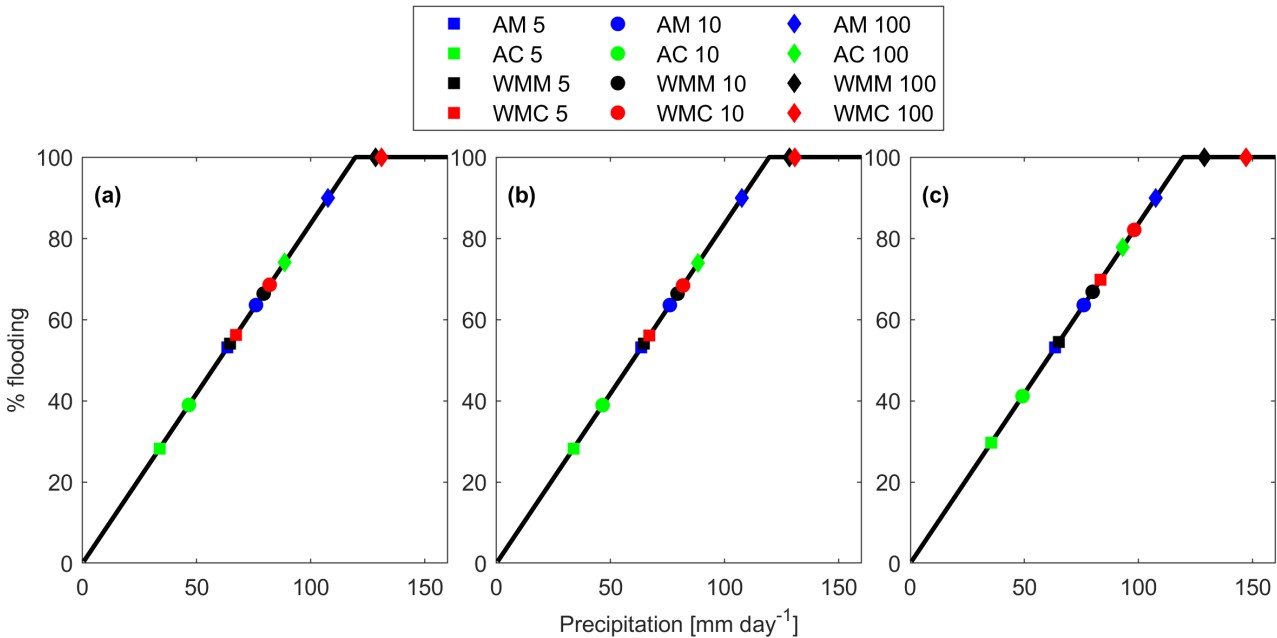

**Figure 12.** Structural scenario 5- (square), 10- (circle), and 100-year (diamond) return periods for AM (blue), AC (green), WMM (black), and WMC (red) data using the (a) Roch-Alegre, (b) Fischer-Kock, and (c) Tawn copulas.



**Table 9.** Precipitation and percent flooding ($\Psi$) associated to the 5-, 10-, and 100-year return periods ($T$) using the Roch-Alegre, Fischer-Kock, and Tawn copulas to determine C1 values. Precipitation values are in $\mathrm{mmday}^{-1}$ and $\Psi$ is a percentage. Values are based off a OWL of $\geq 1.68$ m which forces tide valve closure.

| $T$ | 5-yr. | | 10-yr. | | 100-yr. | |
|---|---|---|---|---|---|---|
| | Precip | $\Psi$ | Precip | $\Psi$ | Precip | $\Psi$ |
| Roch-Alegre | | | | | | |
| AM | 63.51 | 53.14 | 75.97 | 63.56 | 107.44 | 89.89 |
| AC | 33.70 | 28.19 | 46.57 | 38.97 | 88.50 | 74.05 |
| WMM | 64.57 | 54.02 | 79.34 | 66.38 | 128.44 | 100.00 |
| WMC | 67.17 | 56.20 | 81.95 | 68.56 | 130.92 | 100.00 |
| Fischer-Kock | | | | | | |
| AM | 63.51 | 53.14 | 75.97 | 63.56 | 107.44 | 89.89 |
| AC | 33.68 | 28.18 | 46.50 | 38.91 | 88.35 | 73.92 |
| WMM | 64.56 | 54.02 | 79.33 | 66.38 | 128.35 | 100.00 |
| WMC | 66.99 | 56.05 | 81.74 | 68.39 | 130.62 | 100.00 |
| Tawn | | | | | | |
| AM | 63.51 | 53.14 | 75.97 | 63.56 | 107.44 | 89.89 |
| AC | 33.42 | 29.64 | 49.15 | 41.12 | 92.98 | 77.79 |
| WMM | 65.12 | 54.48 | 79.88 | 66.83 | 128.88 | 100.00 |
| WMC | 83.39 | 69.77 | 98.10 | 82.08 | 146.99 | 100.00 |

concentrate probabilities around more centralized intermediate OWL and precipitation values for compound events. This is most pronounced at higher (i.e., 100-year) return periods (Fig. 7). Given that nearly all copulas, save the Cubic, Independent, and Tawn, exhibit similar ML values (for a given sampling) suggests that choosing a reasonable copula may be sufficient to provide a robust characterization of considered compound flooding events.

The choice in sampling imparts a significant influence on flood risk interpretation. Substantially lower ML values are observed in annual sampling (Fig. 4a-f) compared to wet season sampling (Fig. 4 g-i). This likely results from the the increased data availability when the entire wet season is used compared to a single annual observation. When maximum versus coinciding sampling is considered, maximum samplings (AM and WMM) tend to provide the largest OWL at low return periods (Fig. 8a, c, e, Fig. 9, and Table 7). At larger return periods, WMC then provides significantly larger OWLs (Fig. 8a, c, e, Fig. 10, and Table 8). This is observed in the conditionals and bivariates (minus the AND and K hazard scenarios which maximum samplings display the largest OWLs) at all sites. From a logical perspective, coinciding sampling provides a more realistic view of compound events (by definition these are pairs that have co-occurred to produce a compound flooding event). At long return periods AC sampling may require a long data record, which is often unavailable. Notably, in this study AC produced OWL samplings that were substantially lower than any of the other samplings. For example when comparing 100-year OWLs with AC and WMC samplings, the marginal was 32 cm lower and the AND scenario was 17 cm lower (Table 8). Given that



sea wall protected urban coastal areas are highly sensitive to even minor elevations differences (e.g., Gallien et al., 2011), this suggests with limited data records AC sampling should be avoided.

An important note is each probability type appropriately describes a unique event, characterized by OWL and precipitation. Serinaldi (2015) suggests inter-comparing univariate, multivariate, and conditional probabilities and return periods is misleading as each probability type describes its associated event. Events where only extreme OWL or precipitation is of concern, should simply utilize marginal statistics and follow current FEMA guidelines. Compound event analysis may utilize a variety of scenarios. Conditional type distributions become useful when future information on one variable is known (ex., predicted

OWL levels). AND scenarios may be applied when both variables exceeding given limits is of concern. The Survival Kendall scenario is an alternative to the AND scenario using a more conservative approach to develop events of concern (Salvadori et al., 2013). An OR scenario should be applied when either compounding variable exceeding a limit is of concern, whereas the Kendall scenario provides minimum OR events of concern (Salvadori et al., 2011). The benefit of the Kendall and Survival Kendall is all the events along their isolines describe a similar probability space versus the AND and OR isolines describe

events with similar probabilities of non-exceedance. The majority of previous studies focus on specific probability types and do not consider multiple flooding pathways. Only a single study explores all the probabilities associated to different extreme events (Serinaldi, 2016).

From a regulatory perspective, FEMA recommends individual (univariate) analysis to develop return periods for compound coastal flooding applications (FEMA, 2011, 2016c, 2020), and blending the two hazard mapping results. Fundamentally, this

type of approach assumes (event) independence and may underestimate compound flood hazards (e.g., Moftakhari et al., 2019; Muñoz et al., 2020). FEMA provides guidelines for coastal-riverine (FEMA, 2020), tide, surge, tide-surge (FEMA, 2016a), surge-riverine (FEMA, 2016c), and tropical storm (or hurricane ) type flooding events (FEMA, 2016b). Currently, FEMA does not cover compound coastal flooding from high marine water levels and precipitation. However, this work suggests, at high return periods the sampling method is critical to characterizing both univariate and joint probabilities.

Structural scenarios provide a quantitative context to frame flood vulnerability. In the structural failure context, AC sampling significantly underestimates flooding at all return periods, and AM sampling underestimates severe (i.e., 100-year) events, echoing previous AC and AM samplings issues. Similar values between most copulas support the suggestion that choosing a reasonable copula will provide robust results in these types of applications. Precipitation events in the Structural scenario (Table 9) range between $33.70 \, \mathrm{mmday^{-1}}$ and $128.44 \, \mathrm{mmday^{-1}}$, resulting between 28.18 % and complete (100 %) backshore

flooding. This significant flooding at all return periods suggests severe flood vulnerability, which is validated by frequent closures of PCH. This structural function provides a quick and simple alternative to poorly performing bathtub flood models (e.g., Bernatchez et al., 2011; Gallien et al., 2011, 2014; Gallien, 2016) to quantitatively explore flood severity while accounting for infrastructure and joint probabilities.

The maximum OWL and precipitation observations within the record are 2.33 m and $118.93 \, \mathrm{mmday^{-1}}$ for Sunset, 2.27

m and $42.11 \, \mathrm{mmday^{-1}}$ for San Diego, and 2.45 m and $76.83 \, \mathrm{mmday^{-1}}$ for Santa Monica. Most likely precipitation and OWL pairs in high return periods often exceed the current data record's maximums (e.g. Table 8). This study is limited to the available data records and sea level rise clearly imparts a non-stationary trend. Current water level values restricted to today's





distribution tails, will become more frequent in the next century (Taherkhani et al., 2020). For example, Wahl et al. (2015) suggests a previously 100-year event in New York is now a 42-year event based on the increasing correlation between extreme

precipitation and storm surge events. Similarly, our results suggest increasing precipitation and, particularly, OWL levels.

## 6   Conclusions

Univariate, conditional, and bivariate compound flood risk from OWL and/or precipitation were explored at three sites in a tidally dominated, semi-arid region. Seventeen copulas were considered. Previous studies typically relied upon a small number of copulas (e.g. Clayton, Frank, Gumbel, Student t, and Gaussian) for compound flooding assessments. In this case, the

Fischer-Kock, Roch-Alegre, and Tawn copulas produced similar, quality fits across all sampling methods. Although, in some cases, the Tawn produced conservative results. Independent and Cubic copulas consistently produced poor fits which under-/overestimated values and favored either precipitation or water level dominated events. Multiple copulas exhibit similar goodness of fit values (Fig. 4) and most probable pairs (e.g., Fig. 6, 7) suggesting a number of potential copulas may provide a robust multivariate analysis. This work focused solely on the exploring the conditional and joint probabilities of OWL and pre-

cipitation in a tidal and wave dominated semi-arid region. Although wave impacts were not included in this assessment, they are fundamental to coastal flooding, particular in regions subjected to long period swell. Joint probability methods explicitly including wave contributions to compound flood risk characterization are needed.

The annual maximum method is widely recognized for hazard assessments (FEMA, 2011, 2016c), and is common practice in flood risk analysis (e.g., Baratti et al., 2012; Bezak et al., 2014; Wahl et al., 2015). Concerningly, this work suggests that AM

does not characterize the "worse case" scenarios for extreme events. Water levels are substantially underestimated as annual sampling neglects a large portion of observations (Table 8). Generally, maximum samplings produced larger values at minor return periods but significantly underestimated water levels at longer return periods than WMC sampling. Similarly, AC type sampling (Tables 7, 8) grossly underestimate OWLs. Wet season samplings more than quadruple available data records (Table 2) providing additional information about joint events. Further investigation into monthly coinciding and, where appropriate,

water year coinciding are needed to develop optimal sampling strategies for given regional conditions.

*Data availability.* NOAA precipitation data is available for download at https://www.ncei.noaa.gov/metadata/geoportal/rest/metadata/item/ gov.noaa.ncdc:C00313/html#. Tidal data is available for download on NOAA's Tides & Currents wetbsite (https://tidesandcurrents.noaa.gov)

.

*Author contributions.* JL conducted the primary analysis under the guidance and assistance of TG. Both authors wrote and edited the

manuscript. TG conceived of and funded the work.



*Competing interests.* The authors declare no competing interest.

*Acknowledgements.* This work has been supported by the US Coastal Research Program under contract W912HZ-20-200-004, California Department of Parks and Recreation contact number C1670006, the National Science Foundation Graduate Research Fellowship Program grant number DGE-1650604, The National GEM Consortium Fellowship, and the UCLA Cota-Robles Fellowship. The US Coastal Research
Program (USCRP) is administered by the US Army Corps of Engineers® (USACE), Department of Defense. The content of the information provided in this publication does not necessarily reflect the position or the policy of the government, and no official endorsement should be inferred. The authors' acknowledge the USACE and USCRP's support of their effort to strengthen coastal academic programs and address coastal community needs in the United States. Any opinions, findings, conclusions or recommendations expressed in this material are those of the author(s) and do not necessarily reflect the views of the agencies supporting the work. We would like to acknowledge Drs. Yeulwoo
Kim and Nikos Kalligeris for their constructive feedback which strengthened this manuscript.



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
