# Peer review of "Characterizing multivariate coastal flooding events in a semi-arid region: The implications of copula choice, sampling, and infrastructure"

_Natural Hazards and Earth System Sciences, 2021_

## Referee Comment (RC1)

[referee-annotated manuscript omitted]

---

## Referee Comment (RC2)

This study provides overview of risk assessments coastal and pluvial compound floods across Semi-arid/sub-tropical climate types, a transitional regime that blends both hot tropical climate regimes of the South and temperate climate zone. The insight driven from the study helps in developing regional resilience to floods as they have claimed the FEMA provides guidelines for coastal-fluvial, tide-surge, surge-riverine interactions related hazards in addition to the coastal hazards solely-based on tide, surge and hurricane-induced flooding events. The study also provides recommendation of the choice of copulas and sensitivity of hazard to sample types.

Although the added value and motivation for the analyses is robust and seeks immediate attention, the analyzed method has several limitations. Therefore, I would recommend for major revisions before acceptance to this venue. I have summarized my comments as below:

**A. Major Comments**

1. Their definition of compound events are not correct. The sampled AM extremes do not represent the compound event. Since unlike drought, which is a slowly developing phenomena, the occurrence of floods is faster; varies from few hours (*e.g.*, flash flood) up to a week of time scale (considering inundation effects). The authors define AM event to be when the maximum sampling pairs of single largest precipitation and Observed Water Level (OWL) happens within a year or month. However, the paired events can only be qualify as 'compound coastal-pluvial or riverine floods' when the two drivers occurs coincidently if not successively within a limited time window. Based on large-scale climatic pattern this time window is often taken as a within a week of occurrence of the first event. This is because it may take a few days to inundate from rivers as well as coast to know the combined impact, which may not be possible to detect within ± 1-day of occurrence of the event. Secondly, a large watershed may respond within few days of occurrence of the storm event – it may take a few days' time to reach water to flow to the outlet, when it meet with coastal storms. Therefore, a lag-days to be considered to sample such events, however, it would be based on the time of concentration of the watershed and not the user defined input.

Considering a single largest precipitation observation within a year or month to the largest OWL observation within its 24-hour accumulation period, is the one, which could be categorized as a compound event, per *se*. However, here also the question lies whether the watershed is large/medium sized and the land use pattern of the watershed. For example, in case of a large rural (or agricultural intensive) watershed the time of concentration would depend upon the catchment area and flow path length to travel water from the remote point to the catchment outlet. Therefore, the choice of 24-hr/a day may not be adequate to model dependency between two drivers.

Based on the above two comments I find the definition of compound event sampling adopted in the manuscript is erroneous. Rather, their definition could be categorized to solely a multivariate interpretation of extremes.

Further, for wet season monthly maximum and wet season monthly coinciding method of samples, the sampling methods are not properly described. It is not clear whether these two samples follow the iid behavior. This is because as the sampling was performed only based on wet seasons, if they are sampling large number of events then the method fails to preserve iid assumption for frequency analysis. This would calls for nonstationary method, instead of stationary method adopted here.

For example, in Page 24, Line 309: By sampling entire wet season, you would not be able to sample iid events; also it is not the true representative of rare events.

2. The authors have used the vast array of copulas, based on their simulation they infer that the three families are describing the paired event characteristics sufficiently well. However, they have not shown any formal goodness-of-fit that suggests credibility of selected copulas to fit the multivariate extremes. Neither, they discuss out of the three which copula family performs the best for modelling multivariate extremes.

3. Although they have coined a term "Structural" in the return period concept, but in the discussion or results section, I could not find any dedicated section that distinguishes structural or failure probability concepts.

4. 15. Line 327: 'OR' scenario may not provide an accurate estimate of compounding condition since in this case, only one of the variable is assumed to exceed over the other. The simultaneous or joint exceedance of variables are considered in 'AND' scenario case only.

**B. Minor comments**

5. Page 3, line # 60: Although authors have pointed annual maxima sampling generates a worst case scenario; this has been followed earlier. Moftakhari et al. (2017) modeled failure probability for the extreme scenario aiding disaster response considering concurrence of the largest annual freshwater inflow to the lower estuary and the corresponding largest observed hourly water level within ±1 day.

6. Page 3, line # 65: cold & wet fall season (Ganguli et al., 2019a.b), where authors compare coastal compound floods relative to winter seasonal peak discharge and found larger amplifications upon considering compounding effects than solely accounting for high winter (November-March) discharge over northwestern Europe.

7. Page 3, line # 89: The authors have claimed that in current compound flood literature, the terms tides and storm surges are interchanged; this is not true. Check Devlin et al., (2017); Ganguli and Merz, (2019); Ganguli et al. (2020) as a reference. While in Ganguli and Merz (2019) observed high coastal water level was used that includes tides, storm surges and wave setup, for Ganguli et al. (2020) only meteorologically driven skew surge was considered.

8. Table 1: several references are missing:

a. River discharge and water level: (Ganguli and Merz, 2019a, b).
b. River discharge and storm surge: (Ganguli et al., 2020).
c. River discharge and volume (Reddy and Ganguli, 2012).
d. Rainfall and tide: (Bevacqua et al., 2020).
e. Combination of river discharge, volume and duration (Ganguli and Reddy, 2012).

9. Fig. 2 Captions: Expand each of the terms, AM, AC, WMM, WMC, SM, S, and SD.

10. Page 8: Line #148: separates supercritical vs sub-critical region.

11. In Eqn. 12: it was not shown how individual marginal CDFs were included in the expression to get the Kendall 'AND' return period.

12. In Eqn. 13: Are you considering Kendall's return period to define this case? If it is, it should be properly expressed.

13. Page 10: Line 210, the choice of marginal distribution depends on the tail property: if the shape of density function show fast decaying pattern; exponential /Gumbel (GEV-I) distribution would be good; however, for long upper tail, heavier tail distribution is being preferred. Therefore, I would suggest to summarize the basic summary statistics of driver variables, including their skewness and excess kurtosis. Based on that they can make inferences, why certain univariate marginal fits the best.

14. Page 12: line #245, It is not clear if the goodness-of-fit is performed for the choice of copulas? Different copulas behaves differently and one has to select a copula class, which can represent the sample on the basis of their ability to simulate complete vs upper tail dependences. This concern I have also raised as a major comment.

15. Page 18: Line 269, We only consider compounding aspects if they occur within a limited or close time intervals, for example, within a week of occurrence; because of large sized of a catchment, it is physically may not be possible that both events co-occur simultaneously; a lag effect to be considered during event sampling. The response of medium to large sized watershed to a rain event is proportional to the time of concentration of the watershed. This can be utilized to estimate the lag effect.

**References:**

Bevacqua, E., Vousdoukas, M. I., Zappa, G., Hodges, K., Shepherd, T. G., Maraun, D., Mentaschi, L., and Feyen, L.: More meteorological events that drive compound coastal flooding are projected under climate change, Commun Earth Environ, 1, 1–11, https://doi.org/10.1038/s43247-020-00044-z, 2020.

Devlin, A. T., Jay, D. A., Talke, S. A., Zaron, E. D., Pan, J., and Lin, H.: Coupling of sea level and tidal range changes, with implications for future water levels, Scientific reports, 7, 17021, 2017.

Ganguli, P. and Merz, B.: Extreme Coastal Water Levels Exacerbate Fluvial Flood Hazards in Northwestern Europe, Sci Rep, 9, 1–14, https://doi.org/10.1038/s41598-019-49822-6, 2019a.

Ganguli, P. and Merz, B.: Trends in compound flooding in northwestern Europe during 1901–2014, 46, 10810–10820, 2019b.

Ganguli, P. and Reddy, M. J.: Probabilistic assessment of flood risks using trivariate copulas, Theor Appl Climatol, 111, 341–360, https://doi.org/10.1007/s00704-012-0664-4, 2012.

Ganguli, P., Paprotny, D., Hasan, M., Güntner, A., and Merz, B.: Projected Changes in Compound Flood Hazard From Riverine and Coastal Floods in Northwestern Europe, 8, e2020EF001752, https://doi.org/10.1029/2020EF001752, 2020.

Moftakhari, H. R., Salvadori, G., AghaKouchak, A., Sanders, B. F., and Matthew, R. A.: Compounding effects of sea level rise and fluvial flooding, Proceedings of the National Academy of Sciences, 114, 9785–9790, 2017.

Reddy, M. J. and Ganguli, P.: Bivariate flood frequency analysis of upper Godavari river flows using Archimedean copulas, 26, 3995–4018, 2012.

---

## Author Comment (AC1)

The authors would like to thank the reviewers for their constructive comments that have improved this manuscript.

The reviewer's comments (or edits) have been numbered using the format Q.L where Q is the reviewer comment and L is the line number referred to in the reviewer uploaded file "reviewer's nhess-2021-241-RC1.pdf". The authors' response to each comment is given below each question/comment using the format R.L.

**Q.L9** remove ','

**R.L9** The comma has been removed.

**Q.L12** I'm sure this will be explained later, but right now I don't understand what this means.

**R.L12** The data pairs are shown in Table 2 and the authors have updated the text for clarity. The new text reads: Wet season coinciding water level and precipitation pairs benefit from a dramatic increase in data pairs, improved goodness of fit statistics, and provide a range of physically realistic pairs.

**Q.L21** I find this a bit misleading. The fact that such a small SLR causes a doubling of the odds of the 50-year flood event only shows there is a relatively small difference between the 25-year and 50-year event (namely 5 cm), probably be because of relatively modest storm surges. That means Southern California is actually fortunate to not have very extreme high sea levels during extreme events. And that actually makes the area potentially less vulnerable.

**R.L21** The authors agree that typical US West Coast storm surge magnitudes are small (~10 cm, Flick et al., 1998) when compared to multi-meter hurricane generated storm surges experienced in regions with wider continental shelfs. Along the US West Coast, tides dominate marine water levels. Urbanized regions have been built to accommodate the spring tides. Ironically, it is this modest storm surge that make the region highly sensitive to even minor changes in sea level. For example, a 10 cm sea level rise results in spring tides being identical to (or larger than) many historical storm event water levels. The impacts of sea level rise on coastal flooding and vulnerability in California have been demonstrated in the literature (e.g., Tebaldi et al., 2012, Vitousek et al., 2017, Taherkhani et al., 2020). Moftkahari et al., (2015) shows the impact of minor sea level perturbations on flooding (Figure 1). San Francisco, in particular, is highly sensitive to even small (3 cm) increases in sea level (red outline, Figure 1).

[Figure]

Figure 1. Relative vulnerability along U.S. coast to a unified MSL rise. Adapted from Moftakhari et al., (2015).

**Q.L31** I don't understand the second part of this sentence. How can you have different outcomes for an event? An event is single (not plural) so the outcome is single as well. Perhaps you mean different events with the same return period? That would make more sense grammatically. However, in that (multivariate) case it is important to note that the return period loses its meaning (as known in the univariate sense).

**R.L31** The authors would like to thank the reviewer for pointing out this inconsistency in the language. The authors have adopted the reviewer's suggestion and updated the text for clarification. The new text reads "Notably, compound events that share a common return period may produce vastly different flooding outcomes.".

**Q.L35** remove 'potential'

**R.L35** The word 'potential' has been removed.

**Q.L36** The more severe of what? Which things are compared here in this univariate case? Or is this the multivariate case in which each variable is analyzed separately? Then please state this more explicitly.

**R.L36** The authors have updated the text for clarification. The new text reads "For example, FEMA recommends characterizing compound events by developing univariate water level and discharge statistics, modeling each separately, and then adopting the more severe flooding result for transitional areas (FEMA 2011, 2016c)."

**Q.L35** remove "Ironically"

**R.L35** The word ironically has been removed.

**Q.L81** Reduce?

**R.L81** The authors suggestion has been adopted and the text has been updated.

**Q.L87** Extended compared to what? I would like to know the period of observations for the three stations.

**R.L87** Table 2 in the manuscript shows the observation windows used for the study. The full high/low and hourly OWL and precipitation at all sites up to 8/31/21 are provided for the reviewer in the table immediately below.

| Site | Precipitation | | High-Low Tide | | Hourly Tide | |
|---|---|---|---|---|---|---|
| | Start | End | Start | End | Start | End |
| Santa Monica | 7/1/1948 | 12/19/2013 | 8/1/1979 | 8/31/2021 | 11/22/1973 | 8/31/2021 |
| Sunset | 7/1/1948 | 12/1/2012 | 8/1/1979 | 8/31/2021 | 11/28/1923 | 8/31/2021 |
| San Diego | 7/1/1948 | 12/19/2013 | 7/1/1979 | 8/31/2021 | 8/1/1924 | 8/31/2021 |

Using hourly tide data adds over 50 years of additional observed water level records and 31 additional years overlapping precipitation observations for considering compound events at Sunset and San Diego and six years for Santa Monica. The text has been updated to specify the observation windows. The revised text in L84-89 is:

Observed water levels from the Los Angeles (Station ID: 9410660), La Jolla (Station ID: 9410230), and Santa Monica (Station ID: 9410840) tide gauges are available on NOAA's Tides and Currents for daily high-low, hourly, or six-minute intervals (NOAA, 2021 Accessed 2021d). Verified hourly water levels (m

NAVD88) had the longest record length at all three stations and provided an additional 31-years of observations overlapping precipitation data for Los Angeles and La Jolla, and 6-years for Santa Monica. The resulting observations windows are November 22, 1973 to December 19, 2023 for Santa Monica, July 1, 1948 to December 1, 2012 for Sunset and July 1, 1948 to December 19, 2013 for San Diego (Table 2).

**Q.L96** Are. How was this done?

**R.L96** The authors thank the reviewer for catching this grammatical error. The text has been updated with "are". Precipitation measurements were converted to mm/hr by dividing the total event precipitation by the event time. This is reflected in the revised text which now reads "Precipitation measurements were converted to a mm/hr rate by dividing the total event precipitation by the event time to match the hourly OWL measurements."

**Q.L108** remove 'also'

**R.L108** The word 'also' has been removed.

**Q.L110** replace its with their

**R.L110** The sentence has been updated per the reviewer's suggestion and now reads "In the case of coinciding sampling, pairs that had three or more OWL measurements missing within the 24-hour window were manually reviewed and removed if their tidal peak was clearly missing. Specifically, for WMM sampling, months with more than half their observations missing were also reviewed and removed if the tidal peak was missing."

**Q.L132** Are

**R.L132** The authors thank the reviewer for catching this grammatical error, is has been changed to are.

**Q.L150** Kendall scenario.

**R.L150** The text has been updated to include the word scenario.

**Q.L170** But pdfs are very easy to construct from cdfs with numerical approximations.

**R.L170** Uncertainties can be quantified and explored without PDFs, but establishing the most likely events associated to a specific return period requires continuous PDFs . The PDF is generated by taking the derivative of the CDF. Any CDF discontinuities (e.g., in piecewise functions) result in an undefined PDF. Several copulas (e.g., Cuadras-Auge, Shih-Louis, Marshal-Olkin, and Fischer-Hinzmann) employ "min" statements in their CDFs which causes their PDFs (i.e., the derivative of the CDF) to have undefined locations (e.g., Sadegh et al., 2018). Similarly, Raftery, Linear-Spearman, and Cube copulas are piecewise CDF functions with conditional statements, imparting undefined locations in the PDF. In other cases, (e.g., Gaussian, Student-t, and Husler-Reis) distribution functions embedded into the CDF results in a complex partial derivate for the conditional scenarios which are required to establish the most likely event values.

**Q.L171** Therefore it was decided to remove …

**R.L171** The text has been updated in accordance with the reviewer's suggestion and now reads "therefore it was decided to remove those copulas…"

**Q.L172** Equation?

**R.L172** The equation reference has been added after "Conditional 3" to reference the associated equation.

**Q.L173** That surprises me. This should not happen with well chosen values of dx and dy.

**R.L173** The probability space is divided into grid spacings of 0.0005 between 0 and 0.8 and 0.00005 from 0.8 to 1. This high resolution interval is designed to prevent any poor estimations caused by discretization, but in isolated instances negative probabilities occur when a partial derivative is calculated.

**Q.L198** Which probability distributions were used to fit the marginals?

**R.L198** Section 4.1 and Table 3 specify the selected probability distributions for marginal statistics by each site and sampling method. The reference to Figure 3 in L198 was removed to avoid confusion.

**Q.L230** I wouldn't call this 'impacts' as that word has other meanings in flood risk management.

**R.L230** The authors thank the reviewer for pointing this out. The text has been revised and now reads "San Diego WMC conditional CDFs display individual copulas effects (Fig. 5)".

**Q.L233** Mutually consistent?

**R.L233** The text has been updated and now reads "The Roch-Alegre and Fischer-Kock provide very similar results for both precipitation and water level (black and green lines, Fig. 5a, b, e, f)."

**Q.L247** I would like some more elaboration on the number of observations ending up at the upper right side of the isolines. Taking the length of the observation period into account that could provide additional evidence in favor of or against some of the copulas.

**R.L247** We will address this comment in two parts: the first addressing the observations surpassing the isolines and the second addressing the length of observations affecting copula selection.

Critical multivariate events may occur in different regions depending on the hazard type. For example, in multivariate drought studies where the axes are precipitation and soil moisture the critical area representing meteorological drought conditions (i.e., non-exceedance-non-exceedance extremes) lies below the isoline (i.e., Region I). Any data pair above and to the right of the isoline would, in this case be considered 'safe' or events of no concern.

[Figure]

Figure 1, Compound event regions, adapted from Hao et al., (2018).

Conversely, in exceedance-exceedance applications like compound flooding events where, in this study, the axes are precipitation and observed water level the events of interest exist in Region III (Figure 1).

Events above and to the right of the isoline represent more extreme compound events. At the 10 year return period (Figure 2a) a number of exceedance-exceedance events would be expected given the near 70-year observation window. When the return period is more extreme (i.e., 100-year, Figure 2b) pairs on the upper right are minimal or zero for well fit copulas (i.e., Roche-Alegre, Fischer-Kock, Tawn).

[Figure]

Figure 2 (a) 10 year and (b) 100 year return periods for various copulas using the AND scenario. Adapted from Figures 6a and 7a, Lucey and Gallien, in review.

Alternatively, sampling impacts can be considered in regard to observations above the isoline. Annual Maximum (AM, Figure 3 blue x) sampling pairs the single largest precipitation and OWL observations within a given year (without regard to co-occurrence), which are clearly shown as exceedance-exceedance. Similarly, Wet season monthly maximum (WMM) pairs the single largest precipitation and OWL observations within each wet season month. Maximum parings (annual or wet season) do not represent an observed compound event since the pairs did not co-occur, rather were developed from sampling the largest water level and precipitation event which occurred in a given time frame. This maximum sampling is recommended FEMA guidance as a "worse-case scenario" approach (FEMA, 2016). Unsurprisingly, this manifests as a number of pairs occurring in region III (blue x's, black dots).

If historically observed, physically realistic events considered using Annual Coinciding (AC) or the Water Month Coinciding (WMC) data pairs, it becomes apparent in the 100-year return period (Figure 3b) that coinciding events are well described by the isoline. Only one event exceeds the isoline (red arrow).

[Figure]

Figure 3, (a) 10 year return period isolines and (b) 100 year return period for the Fischer-Kock.

Second, considering the length of observations (i.e., the number of events) will influence copula selection (Tong et al., 2015, Sadegh et al., 2017).

Tong et al., (2015) explicitly considers data record length on copula selection, distribution characteristics (mean, standard deviation, skewness, autocorrelation), entropy, goodness of fit (Akaike information criterion, AIC), parameter estimator methods, and return period uncertainty. This study utilizes annual maximum flood peak data between 1893 to 2004 along the Yangtze River in China. Only three single parameter copulas are considered: Clayton, Frank, and Gumbel. In circumstances with minimal data availability (<40-years) the best fitting copula varied between the Frank and Gumbel but evolved to a Frank when record lengths were extended (i.e., 40- to 80-years). Copula fittings were insensitive to time period windows (e.g. period between 1910-1992 vs. 1917-1999 were both well fit by the Frank copula). When the data availability was reduced, distribution characteristics varied. For example, entropy, a measurement of disorder (higher entropy meaning a likelier state) decreases with a shorter data length and longer data records improved AIC values (i.e., minimalized the AIC).

In Sadegh et al., (2017), the Tawn (3-parameter), BB1 (2-parameter), and Burr (1-parameter) copulas are fit to precipitation and soil moisture data given 68-years of monthly (816 pairs), 68-years of annual (68 pairs), and 34-years (34 pairs) of annual observations. The Tawn best described the most data dense observations (68 years of monthly data), followed by the BB1 for the 68 years of annual data, and then the Burr for the 34 years of annual data. In this case, the three parameter Tawn was the only copula able to identify an asymmetric dependence between precipitation and (biased towards) soil moisture apparent in the longer, denser data. Additionally, longer records (i.e., increased data availability) reduced uncertainty along the isolines. Although observational record length implications are beyond the scope of this specific study, it is of great interest and we anticipate future work in this area.

**Q.L251** One word.

**R.L251** The text has been changed to whereas.

**Q.L276** remove 'unique'

**R.L276** The word 'unique' has been removed

**Q.L280** remove 'also'

**R.L280** The word 'also' has been removed.

**Q.L283** Equal to or greater than.

**R.L283** The text has been updated and now reads "A water level equal to or greater than 1.68 m NAVD88 forces valve closures…"

**Q.L297** In our study.

**R.L297** The text has been revised and now reads "Gaussian and Student t copulas were excluded from this study due to their lack of a computationally simple derivative or integral"

**Q.L308** remove 'likely'

**R.L308** 'likely' has been removed

**Q.L349** – 119, 42

**R.L349**– All values in the manuscript are provided with two decimal places to maintain consistency.

**Q.L364** remove 'the'

**R.L364** 'the' has been removed

**Q.L366** Particularly

**R.L366** The reviewer's suggestion has been adopted and the revised text reads "…they are fundamental to coastal flooding, particularly in regions…".

**Q.L373** Records are not quadrupled, you just sample more data from the record

**R.L373** The authors agree with the reviewer and the text has been revised for accuracy. The new sentence is "Wet season sampling quadruples data pairs (Table 2), providing additional historical joint event information".

**References**

FEMA: Guidance for Flood Risk Analysis and Mapping: Coastal Flood Frequency and Extreme Value Analysis, https://www.fema.gov/sites/default/files/2020-02/Coastal_Flood_Frequency_and_Extreme_Value_Analysis_Guidance_Nov_2016.pdf, 2016.

Flick, R. E.: A comparison of California tides, storm surges, and mean sea level during the El Niño winters of 1982-83 and 1997-98, Shore& Beach, 66, 7–11, 1998.

Hao, Z., Singh, V. P., and Hao, F.: Compound extremes in hydroclimatology: a review. Water, 10(6), 718, 2018.

NOAA: Tides & Currents, Online, https://tidesandcurrents.noaa.gov, Accessed 2021.

Sadegh, M., Ragno, E., and AghaKouchak, A.: Multivariate Copula Analysis Toolbox (MvCAT): describing dependence and underlying uncertainty using a Bayesian framework, Water Resources Research, 53, 5166–5183, 2017.

Sadegh, M., Moftakhari, H., Gupta, H. V., Ragno, E., Mazdiyasni, O., Sanders, B., Matthew, R., and AghaKouchak, A.: Multihazard scenarios for analysis of compound extreme events, Geophysical Research Letters, 45, 5470–5480, 2018.

Tong, X., Wang, D., Singh, V. P., Wu, J. C., Chen, X., and Chen, Y. F.: Impact of data length on the uncertainty of hydrological copula modeling, Journal of Hydrologic Engineering, 20(4), 05014019, 2015.

---

## Author Comment (AC2)

The authors would like to thank the reviewer for their comprehensive review and constructive comments that have improved this manuscript.

The reviewer's comments have been numbered using the format QX.Y where Q is the reviewer query, X is the reviewer comment number and Y represents a subdivision of the comment if there are multiple items requiring attention. Responses follow the format of reviewer's "nhess-2021-241-RC2_supplement.pdf". The authors' response to each comment is given below each question/comment using the format RX.Y.

**Q1.1** Their definition of compound events are not correct. The sampled AM extremes do not represent the compound event. Since unlike drought, which is a slowly developing phenomena, the occurrence of floods is faster; varies from few hours (e.g., flash flood) up to a week of time scale (considering inundation effects). The authors define AM event to be when the maximum sampling pairs of single largest precipitation and Observed Water Level (OWL) happens within _a year or month_. However, the paired events can only be qualify as 'compound coastal-pluvial or riverine floods' when the two drivers occurs coincidently if not successively within a _limited time window_.

**R1.1**. The authors agree that the annual maximum method does not represent an observed compound event. However, FEMA, a primary flood regulatory agency in the United States specifically suggests characterizing compound events using the annual maximum pairing (FEMA, 2016). The text has been substantially updated to reflect this key distinction that the reviewer points out. The new text in L103-114 of the revised manuscript is:

Compound flood probabilities are determined with combinations of sampling methods: Annual Maximum (AM), Annual Coinciding (AC), Wet Season Monthly Maximum (WMM), and Wet Season Monthly Coinciding (WMC). AM sampling pairs the single largest precipitation and OWL observations within a given year (without regard to co-occurrence), where AC sampling pairs the single largest precipitation observation within a given year to the largest OWL observation within its 24-hour accumulation period. A summary of each sites' associated gauges, observation windows, and number of pairs is provided in Table 2. Southern California's wet season is defined between October to March and provides a majority of the total annual rainfall (Cayan and Roads, 1984; Conil and Hall, 2006). It is likely for extreme compound events to occur during this period.

Wet season monthly maximum pairs the single largest precipitation and OWL observations within each wet season month. Although strictly speaking maximum parings (annual or wet season) do not technically represent an observed compound event since the co-occurrence of precipitation and water levels follows the FEMA guidance for considering a "worse case scenario" approach (FEMA, 2016c). Wet season monthly coinciding sampling pairs the single largest precipitation observation within each wet season month to the largest OWL observation within its 24-hour accumulation period, providing more realistic pairs compared to maximum sampling.

**Q1.2** Based on large-scale climatic pattern this time window is often taken as a within a week of occurrence of the first event. This is because it may take a few days to inundate from rivers as well as coast to know the combined impact, which may not be possible to detect within ± 1-day of occurrence of the event. Secondly, a large watershed may respond within few days of occurrence of the storm event – it may take a few days' time to reach water to flow to the outlet, when it meet with coastal

storms. Therefore, a lag-days to be considered to sample such events, however, it would be based on the time of concentration of the watershed and not the user defined input.

Considering a single largest precipitation observation within a year or month to the largest OWL observation within its 24-hour accumulation period, is the one, which could be categorized as a compound event, per se. However, here also the question lies whether the watershed is large/medium sized and the land use pattern of the watershed. For example, in case of a large rural (or agricultural intensive) watershed the time of concentration would depend upon the catchment area and flow path length to travel water from the remote point to the catchment outlet. Therefore, the choice of 24-hr/a day may not be adequate to model dependency between two drivers.

Based on the above two comments I find the definition of compound event sampling adopted in the manuscript is erroneous. Rather, their definition could be categorized to solely a multivariate interpretation of extremes.

**R1.2** This study considers the co-occurrence of precipitation and high marine water in a semi-arid, highly urbanized coastal watersheds. The key distinction is that the precipitation component causes pluvial (i.e., surface water flooding from intense rainfall) rather than fluvial (i.e., riverine) flooding and occurs on much shorter time scales. The United State Geological Survey (USGS) and the United States Department of Agriculture (USDA) classify individual watersheds with hydrologic unit codes (HUC). HUC10 officially represent watersheds. The time of concentration ($T_c$) for each HUC watershed were estimated using digital terrain data and the Kirpich (Kirpich, 1940), FAA (1970), and SCS Lag (1975) methods that have been previously applied in southern California coastal watersheds (e.g., Kang et al., 2008, Parker et al., 2019). HUC10 watershed $T_c$ range is ~1-10 hours depending on individual watershed characteristics. All locations are less than 24-hour $T_c$. From literature perspective, storm surge and precipitation 24 hour windows are commonly used (e.g., Wahl et al. 2015, Xu et al., 2014, Xu et al., 2019, Yang et al., 2020). Given both the rapid $T_c$ of these particular watersheds and the literature, the authors believe the 24-hour time window is appropriate.

**Q1.3** Further, for wet season monthly maximum and wet season monthly coinciding method of samples, the sampling methods are not properly described.

**R1.3** The authors have provided additional detail and substantially updated the text to more expansively describe the sampling methods. The new text in L110-114 of the revised manuscript is:

Wet season monthly maximum pairs the single largest precipitation and OWL observations within each wet season month. Although strictly speaking maximum parings (annual or wet season) do not technically represent an observed compound event since the co-occurrence of precipitation and water levels follows the FEMA guidance for considering a "worst case scenario" approach (FEMA, 2016c). Wet season monthly coinciding sampling pairs the single largest precipitation observation within each wet season month to the largest OWL observation within its 24-hour accumulation period, providing more realistic pairs compared to maximum sampling.

**Q1.4** It is not clear whether these two samples follow the iid behavior. This is because as the sampling was performed only based on wet seasons, if they are sampling large number of events then the method fails to preserve iid assumption for frequency analysis. This would calls for nonstationary method, instead of stationary method adopted here.

For example, in Page 24, Line 309: By sampling entire wet season, you would not be able to sample iid events; also it is not the true representative of rare events.

**R1.4** Our assumption for iid behavior is consistent with current literature (Mendoza and Jiménez, 2006, Li et al., 2014, Kapelonis et al., 2015, Shao et al., 2020) considering fast time scale events. These studies ensure iid using a minimum separation window between events on the scale from hours to 5-days. Similarly, in our study the separation window between events is sufficiently large to ensure iid given we sample a single event per year, when using annual sampling, and once per month during the wet season (October to March), when using wet season sampling. Additionally, the events occurring in the watersheds within this study are fast responding and will occur within a 24-hour window (please see response R1.2).

**Q2.1.** The authors have used the vast array of copulas, based on their simulation they infer that the three families are describing the paired event characteristics sufficiently well. However, they have not shown any formal goodness-of-fit that suggests credibility of selected copulas to fit the multivariate extremes. Neither, they discuss out of the three which copula family performs the best for modelling multivariate extremes.

**R2.1** The Bayesian Information Criterion (BIC) and Maximum Likelihood (ML) goodness of fit metrics are provided in section 3.5, equations 16-19. Figure 4 presents the ML values for each tested copula and sampling strategy. ML values describe how well the parameters of an assumed distribution represent a given sample (Chapter 7.5 and 7.6 from DeGroot et. al., 2014), and have been previously used as a goodness of fit metric (e.g., Sadegh et al., 2017, 2018). The authors refrain from explicitly recommending a particular copula since a single 'best' copula did not emerge for given samplings across all three sites. However, the authors point out the most well fit (and poorly fit) copulas and their impacts on return periods (e.g., Tables 5, 6). Akaike Information Criterion (AIC, Figure A3) and BIC (Figure A4) goodness of fit metrics have been added to the Appendix and below.

**Q3.1** Although they have coined a term "Structural" in the return period concept, but in the discussion or results section, I could not find any dedicated section that distinguishes structural or failure probability concepts.

**R3.1** Section 3.2.5 specifically introduces the "Structural" hazard scenario and associated literature. The text has been substantially updated to provide additional context. The revised text (L171-175) now reads:

The "Structural" scenario considers the probability of an output from a structural function, $\Psi(x)$ exceeding a design load or capacity (z) (Salvadori et al., 2016). For example, De Michele et al. (2005) and Volpi and Fiori (2014) used a structural function to evaluate a dam spillway while Salvadori et al. (2015) considers the preliminary design of rubble mound breakwater. In this work, the structural failure function focuses on the question ``what is the probability of a water level forcing tide valve closure and subsequent flooding during a precipitation event?".

**Q4.1** 15. Line 327: 'OR' scenario may not provide an accurate estimate of compounding condition since in this case, only one of the variable is assumed to exceed over the other. The simultaneous or joint exceedance of variables are considered in 'AND' scenario case only.

**R4.1** The authors agree with the reviewer's comment that the 'AND' scenario provides only the joint exceedance of both variables. The 'OR' scenario probability space encompasses three regions: events dominated by the primary variable, events dominated by both variables, and events dominated by the secondary variable (Figure 1, Salvadori et al., 2016). Literature presents multiple studies that have considered the "OR" hazard scenarios (Table 1, Salvadori et al. 2016).

[Figure]

Figure 1 - 'OR' and 'And' Probability spaces. Adapted from Salvadori et al., (2016). All definitions are from Salvadori et al., (2016).

**Q5.1** Page 3, line # 60: Although authors have pointed annual maxima sampling generates a worst case scenario; this has been followed earlier. Moftakhari et al. (2017) modeled failure probability for the extreme scenario aiding disaster response considering concurrence of the largest annual freshwater inflow to the lower estuary and the corresponding largest observed hourly water level within ±1 day.

**R5.1** Moftakhari et al. (2017) explores compound coastal-fluvial (marine-riverine) events. In contrast, our study explores a coastal-pluvial type event where high marine water levels and precipitation co-occur. In both cases indeed, the annual maximum presents a 'worst-case'. However, the results are distinct since the events considered fundamentally differ.

**Q6.1** Page 3, line # 65: cold & wet fall season (Ganguli et al., 2019a.b), where authors compare coastal compound floods relative to winter seasonal peak discharge and found larger amplifications upon considering compounding effects than solely accounting for high winter (November-March) discharge over northwestern Europe.

**R6.1** The authors would like to thank the reviewer for pointing out this work and have added the suggested citations to the text. The definition of "winter" used in Ganguli et. al., 2019a, 2019b does not strictly align with the definition of 'wet season' given the differences in climatology, hydrology, and coastal dynamics between Southern California and Northwestern Europe.

**Q7.1** Page 3, line # 89: The authors have claimed that in current compound flood literature, the terms tides and storm surges are interchanged; this is not true. Check Devlin et al., (2017); Ganguli and Merz,

(2019); Ganguli et al. (2020) as a reference. While in Ganguli and Merz (2019) observed high coastal water level was used that includes tides, storm surges and wave setup, for Ganguli et al. (2020) only meteorologically driven skew surge was considered.

**R7.1** The authors thank the reviewer for pointing this error out. The term 'storm surge' should have been 'water level'. The text has been updated to reflect this change and now reads. "It is worth noting, that within the body of compound flooding literature, the terms tide and water level may be interchanged (e.g., Lian et al., 2013; Xu et al., 2014; Tu et al., 2018; Xu et al., 2019, Yang et al., 2020)."

**Q8.1** Table 1: several references are missing:

a. River discharge and water level: (Ganguli and Merz, 2019a, b).

b. River discharge and storm surge: (Ganguli et al., 2020).

c. River discharge and volume (Reddy and Ganguli, 2012).

d. Rainfall and tide: (Bevacqua et al., 2020).

e. Combination of river discharge, volume, and duration (Ganguli and Reddy, 2013).

**R8.1** The authors thank the reviewer for providing these studies. Ganguli and Merz, (2019a, b) was added to "River discharge and water level". Ganguli et al., (2020) was added to "River discharge and storm surge". Reddy and Ganguli, (2012) and Ganguli and Reddy, (2013) were added to "Combination of river discharge, volume, and duration". Bevacqua et al., (2020) was added to "Rainfall and tide".

**Q9.1** Fig. 2 Captions: Expand each of the terms, AM, AC, WMM, WMC, SM, S, and SD.

**R9.1** The reviewer's suggestion has been adopted and the caption updated accordingly.

**Q10.1** Page 8: Line #148: separates supercritical vs sub-critical region.

**R10.1** The text has been updated to include the reviewer's suggestion and now reads: The "Kendall" (K) scenario highlights an infinite set of OR events that separate the subcritical (i.e., "safe") and supercritical (i.e., "dangerous") statistical regions.

**Q11.1** In Eqn. 12: it was not shown how individual marginal CDFs were included in the expression to get the Kendall 'AND' return period.

**R11.1** Equation 12 is related to the "Survival Kendall". The Survival Kendall scenario (Equation 11) utilizes a "survival copula" which utilizes "survival CDFs" detailed in Section 3.2.2. Additionally, the reader is referred to Salvadori et al. (2013), Salvadori and De Michele (2004), and Serinaldi (2015) for additional information.

**Q12.1** In Eqn. 13: Are you considering Kendall's return period to define this case? If it is, it should be properly expressed.

**R12.1** Equation 13 described the "Structural" scenario (e.g., De Michele et al., 2005, Salvadori et al., 2016). Section 3.2.5 specifically introduces the "Structural" hazard scenario and associated literature. Section 4.4 details the application of the "Structural" scenario which, in this case, follows a Conditional 1 type event. Please also see **R3.1** for additional details.

**Q13.1** Page 10: Line 210, the choice of marginal distribution depends on the tail property: if the shape of density function show fast decaying pattern; exponential /Gumbel (GEV-I) distribution would be good; however, for long upper tail, heavier tail distribution is being preferred. Therefore, I would suggest to summarize the basic summary statistics of driver variables, including their skewness and excess kurtosis. Based on that they can make inferences, why certain univariate marginal fits the best.

**R13.1** Marginal BIC values, which MvCAT uses to select optimal distributions, for OWL (Figure A1) and precipitation (Figure A2) are now provided in the Appendix and, for the reviewer's convenience, below. Additionally, marginal distributions were independently tested using a visual chi square test (e.g., Requena et al., 2013).

**Q14.1** Page 12: line #245, It is not clear if the goodness-of-fit is performed for the choice of copulas? Different copulas behaves differently and one has to select a copula class, which can represent the sample on the basis of their ability to simulate complete vs upper tail dependences. This concern I have also raised as a major comment.

**R14.1** Please see responses R2.1 for a comprehensive explanation of goodness-of-fit metrics.

**Q15.1** Page 18: Line 269, We only consider compounding aspects if they occur within a limited or close time intervals, for example, within a week of occurrence; because of large sized of a catchment, it is physically may not be possible that both events co-occur simultaneously; a lag effect to be considered during event sampling. The response of medium to large sized watershed to a rain event is proportional to the time of concentration of the watershed. This can be utilized to estimate the lag effect.

**R15.1** Precipitation drives a pluvial flooding event on the relatively small urbanized coastal watersheds which exhibit a rapid time of concentration on the order of 1-10 hours. The reviewer is referred to **R1.2** for additional information regarding the watershed time of concentration.

[Figure]

Figure A1 – Marginal OWL BIC values per fitted copula for SM (left column), S (middle column), and SD (right column). Annual Maximum (a), (b), (c); Annual Coinciding (d), (e), (f); Wet Season Monthly Maximum (g), (h), (i); and Wet Season Monthly Coinciding (j), (k), (l). The Y-axis is orientated to display best BIC (top) to worst BIC (bottom).

[Figure]

Figure A2 – Marginal precipitation BIC values per fitted copula for SM (left column), S (middle column), and SD (right column). Annual Maximum (a), (b), (c); Annual Coinciding (d), (e), (f); Wet Season Monthly Maximum (g), (h), (i); and Wet Season Monthly Coinciding (j), (k), (l). The Y-axis is orientated to display best BIC (top) to worst BIC (bottom).

[Figure]

Figure A3 – Copula AIC values per fitted copula for SM (left column), S (middle column), and SD (right column). Annual Maximum (a), (b), (c); Annual Coinciding (d), (e), (f); Wet Season Monthly Maximum (g), (h), (i); and Wet Season Monthly Coinciding (j), (k), (l). The Y-axis is orientated to display best AIC (top) to worst BIC (bottom).

[Figure]

Figure A4 – Copula BIC values per fitted copula for SM (left column), S (middle column), and SD (right column). Annual Maximum (a), (b), (c); Annual Coinciding (d), (e), (f); Wet Season Monthly Maximum (g), (h), (i); and Wet Season Monthly Coinciding (j), (k), (l). The Y-axis is orientated to display best BIC (top) to worst BIC (bottom).

**References**

- Bevacqua, E., Vousdoukas, M. I., Zappa, G., Hodges, K., Shepherd, T. G., Maraun, D., Mentaschi, L., and Feyen, L.: More meteorological events that drive compound coastal flooding are projected under climate change, Communications earth & environment, 1, 1–11, 2020.
- De Michele, C., Salvadori, G., Canossi, M., Petaccia, A., and Rosso, R.: Bivariate statistical approach to check adequacy of dam spillway, Journal of Hydrologic Engineering, 10, 50–57, 2005.
- DeGroot, M. H. and Schervish, M. J.: Probability and Statistics, Pearson Education, 4 edn., 2014.
- Diez, D. M., Barr, C. D., & Cetinskaya-Rundel, M.: Open Intro Statistics (3rd edition)., 2015.
- FEMA: Guidance for Flood Risk Analysis and Mapping: Coastal Flood Frequency and Extreme Value Analysis, https://www.fema.gov/sites/default/files/2020-02/Coastal_Flood_Frequency_and_Extreme_Value_Analysis_Guidance_Nov_2016.pdf, 2016.
- Ganguli, P. and Reddy, M. J.: Probabilistic assessment of flood risks using trivariate copulas, Theoretical and applied climatology, 111, 341–360, 2013.
- Ganguli, P. and Merz, B.: Trends in compound flooding in northwestern Europe during 1901–2014, Geophysical Research Letters, 46, 10 810–10 820, 2019a.
- Ganguli, P. and Merz, B.: Extreme coastal water levels exacerbate fluvial flood hazards in Northwestern Europe, Scientific reports, 9, 1–14, 2019b.
- Ganguli, P., Paprotny, D., Hasan, M., Güntner, A., and Merz, B.: Projected changes in compound flood hazard from riverine and coastal floods in northwestern Europe, Earth's Future, 8, e2020EF001 752, 2020.
- Guirguis, K. J., & Avissar, R.: A precipitation climatology and dataset intercomparison for the western United States. Journal of Hydrometeorology, 9(5), 825-841, 2008.
- Kang, J.H., Kayhanian, M. and Stenstrom, M.K.: Predicting the existence of stormwater first flush from the time of concentration, Water research, 42(1-2), pp.220-228, 2008.
- Kapelonis, Z. G., Gavriliadis, P. N., and Athanassoulis, G. A.: Extreme value analysis of dynamical wave climate projections in the Mediterranean Sea, Procedia Computer Science, 66, 210-219, 2015.
- Kirpich, Z. P.: Time of concentration of small agricultural watersheds, Civil engineering, 10(6), 362, 1940.
- Lian, J., Xu, K., and Ma, C.: Joint impact of rainfall and tidal level on flood risk in a coastal city with a complex river network: a case study of Fuzhou City, China, Hydrology and Earth System Sciences, 17, 679, 2013.
- Li, F., Van Gelder, P. H. A. J. M., Ranasinghe, R. W. M. R. J. B., Callaghan, D. P., and Jongejan, R. B.: Probabilistic modelling of extreme storms along the Dutch coast, Coastal Engineering, 86, 1-13, 2014.
- Leonard, M., Westra, S., Phatak, A., Lambert, M., van den Hurk, B., McInnes, K., Risbey, J., Schuster, S., Jakob, D., and Stafford-Smith, M.: A compound event framework for understanding extreme impacts, Wiley Interdisciplinary Reviews: Climate Change, 5(1), 113-128, 2014.
- Mendoza, E. T., and Jiménez, J. A.: A storm classification based on the beach erosion potential in the Catalonian Coast, In Coastal Dynamics 2005: State of the Practice, pp.1-11, 2006.
- Moftakhari, H. R., Salvadori, G., AghaKouchak, A., Sanders, B. F., and Matthew, R. A.: Compounding effects of sea level rise and fluvial flooding, Proceedings of the National Academy of Sciences, 114, 9785–9790, 2017.

- Parker, S.R., Adams, S.K., Lammers, R.W., Stein, E.D. and Bledsoe, B.P.: Targeted hydrologic model calibration to improve prediction of ecologically-relevant flow metrics, Journal of Hydrology, 573, pp.546-556, 2019.
- Reddy, M. J. and Ganguli, P.: Bivariate flood frequency analysis of upper Godavari River flows using Archimedean copulas, Water Resources Management, 26, 3995–4018, 2012.
- Requena, A. I., Mediero, L., and Garrote, L.: A bivariate return period based on copulas for hydrologic dam design: accounting for reservoir routing in risk estimation, Hydrology and Earth System Sciences, 17(8), 3023-3038, 2013.
- Sadegh, M., Ragno, E., and AghaKouchak, A.: Multivariate Copula Analysis Toolbox (MvCAT): describing dependence and underlying uncertainty using a Bayesian framework, Water Resources Research, 53, 5166–5183, 2017.
- Sadegh, M., Moftakhari, H., Gupta, H. V., Ragno, E., Mazdiyasni, O., Sanders, B., Matthew, R., and AghaKouchak, A.: Multihazard scenarios for analysis of compound extreme events, Geophysical Research Letters, 45, 5470–5480, 2018.
- Salvadori, G. and De Michele, C.: Frequency analysis via copulas: Theoretical aspects and applications to hydrological events, Water re-sources research, 40, 2004.
- Salvadori, G., Durante, F., and De Michele, C.: On the return period and design in a multivariate framework, Hydrology and Earth System Sciences, 15, 3293–3305, 2011.
- Salvadori, G., Durante, F., and De Michele, C.: Multivariate return period calculation via survival functions, Water Resources Research, 49, 2308–2311, 2013.
- Salvadori, G., Durante, F., Tomasicchio, G. R., and D'Alessandro, F.: Practical guidelines for the multivariate assessment of the structural risk in coastal and off-shore engineering, Coastal Engineering, 95, 77-83, 2015.
- Salvadori, G., Durante, F., De Michele, C., Bernardi, M., and Petrella, L.: A multivariate copula-based framework for dealing with hazard scenarios and failure probabilities, Water Resources Research, 52, 3701–3721, 2016.
- Seneviratne, S., Nicholls, N., Easterling, D., Goodess, C., Kanae, S., Kossin, J., Luo, Y., Marengo, J., McInnes, K., Rahimi, M., Reichstein, M., Sorteberg, A., Vera, C., and Zhang, X.: Changes in climate extremes and their impacts on the natural physical environment, in: Managing the Risks of Extreme Events and Disasters to Advance Climate Change Adaptation: A Special Report of Working Groups I and II of the Intergovernmental Panel on Climate Change, edited by Field, C., Barros, V., Stocker, T., Qin, D., Dokken, D., Ebi, K., Mastrandrea, M., Mach, K., Plattner, G.-K., Allen, S., Tignor, M., and Midgley, P., pp. 109–230, Cambridge University Press, Cambridge, UK, and New York, NY, USA, 2012.
- Serinaldi, F.: Dismissing return periods!, Stochastic Environmental Research and Risk Assessment, 29, 1179–1189, 2015.
- Shao, Z., Liang, B., and Gao, H.: Extracting independent and identically distributed samples from time series significant wave heights in the Yellow Sea, Coastal Engineering, 158, 103693, 2020.
- Shumway, R. H. and Stoffer, D. S.: Time series analysis and its applications, 3, New York: springer, 2000.
- Tu, X., Du, Y., Singh, V. P., and Chen, X.: Joint distribution of design precipitation and tide and impact of sampling in a coastal area, International Journal of Climatology, 38, e290–e302, 2018.
- U.S. Geological Survey (USGS) and U.S. Department of Agriculture (USDA), Natural Resources Conservation Service, Federal Standards and Procedures for the National Watershed Boundary

Dataset (WBD) (4 ed.): Techniques and Methods 11–A3, 63 p., https://pubs.usgs.gov/tm/11/a3/., 2013.

- Volpi, E. and Fiori, A.: Hydraulic structures subject to bivariate hydrological loads: Return period, design, and risk assessment, Water Resources Research, 50(2), 885-897, 2014.
- Wahl, T., Jain, S., Bender, J., Meyers, S. D., and Luther, M. E.: Increasing risk of compound flooding from storm surge and rainfall for major US cities, Nature Climate Change, 5, 1093, 2015.
- Xu, H., Xu, K., Lian, J., and Ma, C.: Compound effects of rainfall and storm tides on coastal flooding risk, Stochastic Environmental Research and Risk Assessment, 33, 1249–1261, 2019.
- Xu, K., Ma, C., Lian, J., and Bin, L.: Joint probability analysis of extreme precipitation and storm tide in a coastal city under changing environment, PLoS One, 9, e109 341, 2014.
- Yang, X., Wang, J., and Weng, S.: Joint Probability Study of Destructive Factors Related to the "Triad" Phenomenon during Typhoon Events in the Coastal Regions: Taking Jiangsu Province as an Example, Journal of Hydrologic Engineering, 25, 05020 038, 2020.
- Zscheischler, J., Westra, S., Van Den Hurk, B.J., Seneviratne, S.I., Ward, P.J., Pitman, A., AghaKouchak, A., Bresch, D.N., Leonard, M., Wahl, T., and Zhang, X.: Future climate risk from compound events, Nature Climate Change, 8(6), 469-477, 2018.

---

## Referee Report (RR1)

The authors use observations from tide and precipitation gauges at three sites along the West coast of the United States to calculate various joint and conditional probabilities based on different sampling methods. They compare different scenarios of bivariate return periods as defined in Salvadori et al. (2016) and Serinaldi (2015) using a wide range of copulas and marginal distributions. Based on this, they conclude that annual maxima does not produce "worst-case" events and recommend instead wet season coinciding sampling to characterize compound flooding in semi-arid regions.

The authors have done a substantial amount of interesting work: they have tested many different scenarios and possibilities of copulas and marginal distributions. However, I find the manuscript difficult to follow, which as a result, makes the relevance of the findings unclear. More particularly, I find the abstract quite disconnected to the structure and the findings presented in the manuscript. For example, I am still uncertain which research gaps are being filled. I do think this might be resolved by carefully restructuring the manuscript and their findings to focus on the main points the authors are trying to convey.

- As pointed out by the authors, "Serinaldi (2015) suggests inter-comparing univariate, multivariate, and conditional probabilities and return periods is misleading as each probability type describes its associated event." (line 337-338). Yet, after reading the manuscript it gives the impression that this is exactly what the authors did by listing the results obtained from all possible hazard scenarios. In the methodology, the authors mention the type of question each hazard scenario might be answering, but this is not reflected in the abstract. Instead, only a broad mention of "compound events" is mentioned.

- The authors also frequently refer to the FEMA (2016) methodology which seems to imply that they want to compare their hazard scenarios with FEMA but this is not done at any point in the manuscript. FEMA (2016) provides a methodology to derive the 100 (500)- year return period of the water level while this study does not model the obtained water levels from the event pairs obtained. Furthermore, the FEMA flood maps should represent, in theory, the flood depths levels happening once every 100 (or 500) years on average. This probability cannot be compared with probabilities from the hazard scenarios and conditional probabilities as each of them only contains a subset of all potential flood events.

- It is unclear what is meant by "worst-case" in the manuscript which makes the abstract and conclusions very difficult to understand. Based on which results do the authors conclude that AM does not result in worst case scenarios? Is this for each hazard scenario?

- To my understanding, this study does not assess flood risk but derives the pairs of different joint/conditional return periods, except for the structural scenario where a mention on the flood extent is given. As such, I find the title a bit misleading in its current form. Also, the *implications* of the work could be better highlighted in the abstract. I find parts of the discussion interesting (lines 339-348) where the authors mention the value of testing different hazard scenarios to capture compound interactions.

- Line 9-10: "Although annual maximum sampling is commonly recommended for characterizing compound events, …". The current emphasis in the abstract about the annual

maximum sampling is not straightforward to me. This could also be because I am not sure what the authors mean by "worst case" event pairs. But I would actually tend to disagree with this statement. There is a wide variety of sampling methods commonly used to characterize compound floods that include both peaks-over-threshold and annual maxima, with no clear scientific consensus for one or the other method. Compound flood studies that have looked at the impact from these events, such as Santos et al. (2020) found AM methods to produce a more balanced datasets. However, without looking at the impact from your events, one cannot make a strong conclusion about this.

- I find the comparison of different sampling methods particularly confusing (section 4.3 and 4.4). As mentioned by reviewer 2, these methods sample different sets. Therefore, it is expected to lead to different values for the evaluation of the '100-year' probability. When presenting the results, this should be clearly discussed, otherwise this seems to imply that '100-year' event from the marginal, conditional and joint probabilities are the same, which is not true. Maybe this could already be clearly stated in the Methods and reiterated in the Results. In the end, an important message from your study is specifically that these hazard scenarios do not represent the same probability space and thus for the same probability level, they lead to different values. This means that extreme caution should be used for users/modelers who want to use a given hazard scenario to derive the "100-year" flood event.

- The authors did not show formal goodness-of-fit test (but only informal ones like the BIC and AIC). The lowest BIC or AIC value could still denote a poor fit from the model. It is therefore usually recommended to perform the Cramer-von Mises blanket test, see for example:
  - Ward et al. (2018). *Dependence between high sea-level and high river discharge increases flood hazard in global deltas and estuaries*. Doi: 10.1088/1748-9326/aad400
  - Genest et al. (2009). Goodness-of-fit tests for copulas: A review and a power study. Doi: 10.1016/j.insmatheco.2007.10.005
  - Couasnon et al. (2018). A Copula-based bayesian network for modeling compound flood hazard from riverine and coastal interactions at the catchment scale: An application to the Houston ship channel, Texas. Doi: 10.3390/w10091190

Minor comments
The line numbers below refer to the track changes manuscript
- Line 1: Your study assumes stationary conditions so I am not sure the sentence on sea level rise is needed.
- Line 38: "and then adopting the more severe flooding result". From the reference cited (FEMA 2016c), it is shown that FEMA does not uses the most severe flood result but a combination of both fluvial and coastal flood depth probabilities to estimate the 100-year water level in transitional areas (see Figure 4.1 in the *FEMA Guidance for Flood Risk Analysis and Mapping: Coastal Flood Frequency and Extreme Value Analysis*). So I would rephrase this to better highlight the limitations from their methods.

- Line 50-51: Maybe rephrase to mention that *initial* studies were mainly focusing on hazard scenarios. I would then advise to change the title of Table 1 to a non-exhaustive list of studies that used hazard scenarios (if this is what the authors want to emphasize). Otherwise, the body of literature that used copulas to study compound flooding is much broader that what is stated there. Also, there is a lot of recent research that moved beyond single hazard scenarios in order to better capture all potential flood events using copulas. See for example:
    - Bevacqua et al. (2017). Multivariate Statistical Modelling of Compound Events via Pair-Copula Constructions: Analysis of Floods in Ravenna. Doi: 10.5194/hess-21-2701-2017
    - Santos et al. (2021). Assessing compound flooding potential with multivariate statistical models in a complex estuarine system under data constraints. Doi: 10.1111/jfr3.12749
    - Jane et al. (2022). Assessing the Potential for Compound Storm Surge and Extreme River Discharge Events at the Catchment Scale with Statistical Models: Sensitivity Analysis and Recommendations for Best Practice
    - Couasnon et al. (2022). A flood risk framework capturing the seasonality of and dependence between rainfall and sea levels – an application to Ho Chi Minh City, Vietnam. Doi 10.1029/2021WR030002
- Line 67: "humid climatic conditions": This sounds very broad
- Line 75 : "most likely". At this point, this term has not been introduced in the manuscript and without context, this may lead to the wrong understanding of this expression.
- Line 92: "December 19, 2023" Typo?
- Line 93-94: I am not sure of the added value of this sentence. Can you make a clearer link with your study design?
- Line 103: "by the event time". Do you mean duration? Or time step?
- Line 114-116: ".. since the co-occurrence of precipitation and water levels follows the FEMA guidance for considering a "worst case scenario" approach ." I do not understand this. Can you elaborate?
- The authors use a lot of acronyms. I can understand why but the references to the acronym is not always mention in the Figures or titles. So the reader sometimes has to scroll to the text to find those acronyms again. For example, I couldn't find what M refers to (I think Marginal distribution?). I would carefully review all Figures and Tables to make sure any acronym is mentioned to be able to interpret the figure or table independently of the text.
- Table 3 : The word "marginal" or "univariate" when describing the probabilities/return period is missing in the title.
- Line 194: "easily". I would remove this term and rename this section "Univariate return periods". In multivariate space, I think to interpret a return period is not easy at all!
- Line 320: "save the Cubic": typo?
- Uncertainties are not quantified in your study which limits the interpretation of the implications. I am not asking to add this but at least mention this point somewhere in the discussion.

- Similarly, it would be interesting to mention the limitation of this method in locations with multiple pluvial or coastal flood seasons.

---

## Author Response (AR2)

The authors would like to thank the reviewer for their constructive comments that have strengthened this manuscript.

The reviewer's comments have been numbered using the format QX.Y where Q is the reviewer query, X is the reviewer comment number and Y represents a subdivision of the comment if there are multiple items requiring attention. The authors' response to each comment is given below each question/comment using the format RX.Y.

**Q1.1** Their definition of the compound event is wrong. The authors have sampled the largest OWL and the largest precipitation event within a given year without considering a short/limited time window. Typically for compound floods, it is assumed that the time between each driver is from 0 to a few days, preferably within a week to account for synoptic circulation. Since the 'time' of elapsing of one event to another is sufficiently large here – it merely resulted in a bivariate/multivariate event series rather than a compound pair. Likewise, pairing within a wet season month does not contribute towards compound event construction.

**R1.1** The authors refer to Senevirante et al. (2012), Leonard et al. (2014), and Zscheischler et al. (2018) for the definition of a compound event. Coinciding sampling would represent compound events while, the maximum sampling strategy would not. In order to accommodate both event sampling types, the authors have adopted the reviewer's suggestion and updated the manuscript to reflect "multivariate" rather than compound events.

**Q1.2** Although they quote a certain publication from FEMA pointing 'observations within a wet season' as worst possible pairs - In fact, FEMA (2015, 2018) do not provide any unified guidelines for analyzing compound floods. In FEMA (2020), a recommendation for combined occurrence for coastal and riverine flooding is provided, however, no sampling methods were discussed. This is in reference to Lines 110 to 115 in the manuscript.

**R1.2** The authors have substantially updated the text to provide additional context on the sampling and have removed any reference to "worst-case" or worst possible pairs. The new paragraph reads:

Multivariate flood probabilities are determined with combinations of sampling methods: Annual Maximum (AM), Annual Coinciding (AC), Wet Season Monthly Maximum (WMM), and Wet Season Monthly Coinciding (WMC). AM sampling pairs the single largest precipitation and OWL observations within a given year (without regard to co-occurrence), where AC sampling pairs the single largest precipitation observation within a given year to the largest OWL observation within its 24-hour accumulation period. Each sampling method samples from a unique probability space and therefore will provide varying perspectives for a return period. A summary of each sites' associated gauges, observation windows, and number of pairs is provided in Table 2. Southern California's wet season is defined between October to March and provides a majority of the total annual rainfall (Cayan and Roads, 1984; Conil and Hall, 2006). It is likely for extreme multivariate events to occur during this period. Maximum sampling pairs the single largest precipitation and OWL observations within each year or wet season month. A multivariate event created with the largest observed precipitation and OWL within a year or wet season month can result in an event with severe flooding potential. Although strictly speaking maximum parings (annual or wet season) do not technically represent an observed

multivariate event, they would represent a severe event and are consistent with the blended approach recommended by FEMA (2016c). Coinciding sampling pairs the single largest precipitation observation within each year or wet season month to the largest OWL observation within its 24-hour accumulation period, providing more realistic pairs compared to maximum sampling.

**Q2.1** The authors have used AIC and ML methods as a goodness-of-fit estimate to select copula. However, the AIC and ML method does not provide any goodness-of-fit statistics of the population fit and the power of the test - it compares the empirical vs theoretical fit of the data. For the selection of copulas, a double parametric bootstrap method is to be adopted, other than the AIC-based criteria (Genest et al., 2009; Rémillard, 2017).

**R2.1** The authors have performed the analysis using a Cramér-von Mises tests (e.g., Genest et al., 2009, Couasnon et al, 2018, Sadegh et al., 2018, Ward et al., 2018). The manuscript has been updated and revised accordingly.

**Q3.1** In compound event perspective, it is of interest to analyze when the design level exceeds both variables simultaneously, and not either of the variables in isolation. Therefore, the 'OR' criteria is suitable to analyze multivariate probability and not the compound extreme per se. The papers they have followed, also emphasize multivariate events and estimation of associated hazard and therefore, do not explicitly consider compound events in particular.

**R3.1** The authors agree and have revised the manuscript to reflect "multivariate event". Please also see response **R1.1**.

**Definitions of a compound event**

- "(1) two or more extreme events occurring simultaneously or successively; (2) combinations of extreme events with underlying conditions that amplify the impact of the events; or (3) combinations of events that are not themselves extremes but lead to an extreme event or impact when combined. The contributing events can be of similar (clustered multiple events) or different type(s)" (IPCC 2012, Senevirante et al. 2012)
- The combination of processes (climate drivers and hazards) leading to a significant impact is referred to as a 'compound event'. (Zscheischler et al., 2018)
- The combination of variables or events that lead to an extreme impact is referred to as a compound event. (Leonard et al., 2014)

**References**

- Couasnon, A., Sebastian, A., and Morales-Nápoles, O.: A copula-based Bayesian network for modeling compound flood hazard from riverine and coastal interactions at the catchment scale: An application to the Houston Ship Channel, Texas, Water, 10, 1190, 2018.
- FEMA: Guidance for Flood Risk Analysis and Mapping: Coastal Flood Frequency and Extreme Value Analysis, https://www.fema.gov/sites/default/files/2020-02/Coastal_Flood_Frequency_and_Extreme_Value_Analysis_Guidance_Nov_2016.pdf, 2016.
- Genest, C., Rémillard, B., and Beaudoin, D.: Goodness-of-fit tests for copulas: A review and a power study, Insurance: Mathematics and economics, 44, 199–213, 2009.

- Leonard, M., Westra, S., Phatak, A., Lambert, M., van den Hurk, B., McInnes, K., Risbey, J., Schuster, S., Jakob, D., and Stafford-Smith, M.: A compound event framework for understanding extreme impacts, Wiley Interdisciplinary Reviews: Climate Change, 5(1), 113-128, 2014.
- Sadegh, M., Moftakhari, H., Gupta, H. V., Ragno, E., Mazdiyasni, O., Sanders, B., Matthew, R., and AghaKouchak, A.: Multihazard scenarios for analysis of compound extreme events, Geophysical Research Letters, 45, 5470–5480, 2018.
- Ward, P. J., Couasnon, A., Eilander, D., Haigh, I. D., Hendry, A., Muis, S., Veldkamp, T. I., Winsemius, H. C., and Wahl, T.: Dependence between high sea-level and high river discharge increases flood hazard in global deltas and estuaries, Environmental Research Letters, 13, 084 012, 2018.
- Zscheischler, J., Westra, S., Van Den Hurk, B.J., Seneviratne, S.I., Ward, P.J., Pitman, A., AghaKouchak, A., Bresch, D.N., Leonard, M., Wahl, T., and Zhang, X.: Future climate risk from compound events, Nature Climate Change, 8(6), 469-477, 2018.

**Reviewer 2**

The authors would like to thank the reviewer for their constructive comments that have strengthened this manuscript.

The reviewer's comments have been numbered using the format QX.Y where Q is the reviewer query, X is the reviewer comment number and Y represents a subdivision of the comment if there are multiple items requiring attention. The authors' response to each comment is given below each question/comment using the format RX.Y. Comments are address in the order they appear from the reviewer.

**Q1.1** As pointed out by the authors, "Serinaldi (2015) suggests inter-comparing univariate, multivariate, and conditional probabilities and return periods is misleading as each probability type describes its associated event." (line 337-338). Yet, after reading the manuscript it gives the impression that this is exactly what the authors did by listing the results obtained from all possible hazard scenarios. In the methodology, the authors mention the type of question each hazard scenario might be answering, but this is not reflected in the abstract. Instead, only a broad mention of "compound events" is mentioned.

**R1.1** The abstract and manuscript have been substantially updated in accordance with the reviewer's suggestion. The abstract now specifically refers to hazard scenarios (Line 6). Lines 327-335 clarify when and how each scenario may be applied. Similarly, additional context was added at Lines 335-337 to emphasize the purpose of exploring all possible hazard scenarios within this study. Please refer to the redline text for the associated updates.

**Q2.1** The authors also frequently refer to the FEMA (2016) methodology which seems to imply that they want to compare their hazard scenarios with FEMA but this is not done at any point in the manuscript. FEMA (2016) provides a methodology to derive the 100 (500)- year return period of the water level while this study does not model the obtained water levels from the event pairs obtained. Furthermore, the FEMA flood maps should represent, in theory, the flood depths levels happening once every 100 (or 500) years on average. This probability cannot be compared with probabilities from the hazard scenarios and conditional probabilities as each of them only contains a subset of all potential flood events.

**R2.1** The currently recommended FEMA methodology represents a simple and efficient methodology relying on annual maximum or peaks over threshold sampling to consider flood hazards. Nonlinear interactions may amplify flood risk in areas where multi-source flooding occurs (e.g., Moftakhari et al., 2019). Currently, FEMA does not provide guidance on methods estimating flood risks associated with the co-occurrence of precipitation and high water levels. This study explores the influence of sampling and copula selection on precipitation and water level pairs. Although it is outside the scope of work presented here, future work will hydrodynamically model precipitation-water level pairs and quantitatively compare flood extent to univariate, blended models similar to the FEMA methodology. The manuscript has been edited to avoid any suggestion that these hazard scenarios were hydrodynamically modeled and directly compared to FEMA flood maps.

**Q3.1** It is unclear what is meant by "worst-case" in the manuscript which makes the abstract and conclusions very difficult to understand. Based on which results do the authors conclude that AM does not result in worst case scenarios? Is this for each hazard scenario?

**R3.1** All references to "worst-case" has been removed in the manuscript. Annual maximum sampling pairs the largest observations within a year creating a data set of multivariate events with relatively large OWL and precipitation. Generally, the annual maximum sampling provides the largest OWL values. However, the 100-year return period for OR and SK scenarios exhibited larger OWL values using water year monthly coinciding sampling (WMC; Figure 9b, c) as discussed in the paragraph starting on Line 269. The paragraph stating on Line 269 in the manuscript has been expanded and revised to provide additional context.

[Figure]

**Figure 9.** San Diego (a) AND, (b) OR, (c) SK, and (d) K hazard scenarios for annual maximum (AM, cross), annual coinciding (AC, plus), wet season monthly maximum (WMM, dot), wet season monthly coinciding (WMC, triangle) data and 100-year isolines using the BB1 copula. Sampling labels point to the mostly likely value on their respective isolines.

This behavior was also observed in the marginal and conditional cases (paragraph starting Line 257) and led to the conclusion that annual maximum sampling may not result in the largest potential flood events at long return periods. Additional text was added to provide further clarification to Line 276 which reads "Given wet season monthly coinciding sampling results in larger OWL values for the marginal, conditional, OR, and Kendall scenarios, this suggests maximum type sampling may not accurately reflect OWL at extreme return periods".

**Q4.1** To my understanding, this study does not assess flood risk but derives the pairs of different joint/conditional return periods, except for the structural scenario where a mention on the flood extent is given. As such, I find the title a bit misleading in its current form. Also, the implications of the work could be better highlighted in the abstract. I find parts of the discussion interesting (lines 339-348)

where the authors mention the value of testing different hazard scenarios to capture compound interactions.

**R4.1** The authors have amended the text to consider events rather than flood risk. The title has been updated to reflect the reviewer's suggestion and is now "Characterizing multivariate coastal flooding events in a semi-arid region: The implications of copula choice, sampling, and infrastructure". The abstract has been updated

**Q5.1** Line 9-10: "Although annual maximum sampling is commonly recommended for characterizing compound events, …". The current emphasis in the abstract about the annual maximum sampling is not straightforward to me. This could also be because I am not sure what the authors mean by "worst case" event pairs. But I would actually tend to disagree with this statement. There is a wide variety of sampling methods commonly used to characterize compound floods that include both peaks-over-threshold and annual maxima, with no clear scientific consensus for one or the other method. Compound flood studies that have looked at the impact from these events, such as Santos et al. (2020) found AM methods to produce a more balanced datasets. However, without looking at the impact from your events, one cannot make a strong conclusion about this.

**R5.1** The authors have substantially updated the abstract and manuscript to remove any references to "worst-case" and provide a more balanced approach to potential sampling methods. Please refer to the redline text for changes. The authors agree there is not a scientific consensus for one sampling method or the other and provide multiple studies utilizing both methods in the paragraph starting on Line 50. "Recommended" has been changed to "used" to prevent suggestion of any particular sampling method. Santos et al., (2021) recommends monthly maximum for more balanced datasets. In this case, the monthly maximum paring exhibited inverse relationships owing to the dry season where essentially no precipitation occurs. As an alternative, results are presented for the wet season monthly maximum.

**Q6.1** I find the comparison of different sampling methods particularly confusing (section 4.3 and 4.4). As mentioned by reviewer 2, these methods sample different sets. Therefore, it is expected to lead to different values for the evaluation of the '100-year' probability. When presenting the results, this should be clearly discussed, otherwise this seems to imply that '100-year' event from the marginal, conditional and joint probabilities are the same, which is not true. Maybe this could already be clearly stated in the Methods and reiterated in the Results. In the end, an important message from your study is specifically that these hazard scenarios do not represent the same probability space and thus for the same probability level, they lead to different values. This means that extreme caution should be used for users/modelers who want to use a given hazard scenario to derive the "100-year" flood event.

**R6.1** The authors agree that the various sampling methods represent different probability spaces and should result in different return period values. The variation among sampling spaces has yet to be explored within the literature. Text was added on Line 109 and Line 257 to provide context and clarify this point. The added text reads: "Each sampling method samples from a unique probability space and therefore will provide varying perspectives for a return period" and "It should be noted that each sampling method represents a unique probability space and accordingly results in alternative realizations of a given return period. ".

**Q7.1** The authors did not show formal goodness-of-fit test (but only informal ones like the BIC and AIC). The lowest BIC or AIC value could still denote a poor fit from the model. It is therefore usually recommended to perform the Cramer-von Mises blanket test.

**R7.1** The authors have updated the analysis using the suggested Cramér-von Mises tests (e.g., Genest et al., 2009, Couasnon et al, 2018, Sadegh et al., 2018, Ward et al., 2018), and the manuscript updated accordingly. Although the specific copula selections shifted, the overall findings are similar. Please refer to the updated Methods, Results, and Discussion sections within the redline manuscript to see the respective changes.

**Q8.1** Line 1: Your study assumes stationary conditions so I am not sure the sentence on sea level rise is needed.

**R8.1** The sentence was removed per the reviewer's suggestion.

**Q9.1** Line 38: "and then adopting the more severe flooding result". From the reference cited (FEMA 2016c), it is shown that FEMA does not uses the most severe flood result but a combination of both fluvial and coastal flood depth probabilities to estimate the 100-year water level in transitional areas (see Figure 4.1 in the FEMA Guidance for Flood Risk Analysis and Mapping: Coastal Flood Frequency and Extreme Value Analysis). So I would rephrase this to better highlight the limitations from their methods.

**R9.1** The authors thank the reviewer for pointing out this error. Lines 35 to 37 have been amended and now read, "For example, FEMA recommends characterizing multivariate events by developing univariate water level and discharge statistics and then adopting a smoothed, blended result for transitional areas (FEMA, 2011, 2016c)."

**Q10.1** Line 50-51: Maybe rephrase to mention that initial studies were mainly focusing on hazard scenarios. I would then advise to change the title of Table 1 to a non-exhaustive list of studies that used hazard scenarios (if this is what the authors want to emphasize). Otherwise, the body of literature that used copulas to study compound flooding is much broader that what is stated there. Also, there is a lot of recent research that moved beyond single hazard scenarios in order to better capture all potential flood events using copulas.

**R10.1** The reviewer's suggestion was adopted. Line 58 in the updated manuscript now reads as "Initial studies were primarily focused on select hazard scenarios…" and Table 1's caption will now start as, "A non-exhaustive list of multivariate studies…", per the reviewer's comments.

**Q11.1** Line 67: "humid climatic conditions": This sounds very broad

**R11.1** The authors have added specific information to constrain the term "humid climatic conditions". Line 67 now includes a reference to Koppen-Geiger system, Beck et al. (2018).

**Q12.1** Line 75 : "most likely". At this point, this term has not been introduced in the manuscript and without context, this may lead to the wrong understanding of this expression.

**R12.1** The authors thank the reviewer for pointing out this subtle but important point. The text has been updated to remove "most likely" and replace with "resulting".

**Q13.1** Line 92: "December 19, 2023" Typo?

**R13.1** The authors thank the reviewer for catching this typo and changed "2023" to "2013".

**Q14.1** Line 93-94: I am not sure of the added value of this sentence. Can you make a clearer link with your study design?

**R14.1** From an oceanographic prospective, the term tide specifies water levels changes only from astronomical influences. Observed water level represents tide and any storm surge (i.e., inverse barometric effects and wind/wave setup) or climatic influences (such as El Nino). A sentence was added at Line 92 to clarify the importance of the distinction between observed water level and tide. The additional text reads: "This is a key distinction since compound event dependencies may change depending on water level selection."

**Q15.1** Line 103: "by the event time". Do you mean duration? Or time step?

**R15.1** "Event time" has been updated to "event duration" in accordance with the reviewer's suggestion.

**Q16.1** Line 114-116: ".. since the co-occurrence of precipitation and water levels follows the FEMA guidance for considering a "worst case scenario" approach ." I do not understand this. Can you elaborate?

**R16.1** The language "worst-case" has been removed and new text added to Line 115. The added text reads "Although strictly speaking maximum parings (annual or wet season) do not technically represent an observed multivariate event, they would represent a severe event and are consistent with the blended approach recommended by FEMA (2016)".

**Q17.1** The authors use a lot of acronyms. I can understand why but the references to the acronym is not always mention in the Figures or titles. So the reader sometimes has to scroll to the text to find those acronyms again. For example, I couldn't find what M refers to (I think Marginal distribution?). I would carefully review all Figures and Tables to make sure any acronym is mentioned to be able to interpret the figure or table independently of the text.

**R17.1** All tables and figures have updated captions that provide utilized acronyms.

**Q18.1** Table 3: The word "marginal" or "univariate" when describing the probabilities/return period is missing in the title.

**R18.1** The probabilities in Table 3 are for all scenarios (marginal, conditional, bivariate) utilizing Wet Season Monthly Maximum or Wet Season Monthly Coinciding samplings. Probabilities of exceedance (or their inverse, the return period) describe a probability threshold or front (in the bivariate case) dependent on sampling (Eq. 15 in the manuscript) and the available data (length, observations, spread, etc.). The return period is not unique to the marginal case, and therefore not specified as such. Table 4 provides the fitted marginal distributions and Figure 3 displays the marginal CDFs.

**Q19.1** Line 194: "easily". I would remove this term and rename this section "Univariate return periods". In multivariate space, I think to interpret a return period is not easy at all!

**R19.1** The term "easily" was removed and the sentence now reads "Return periods (T) provide a metric describing the severity of an event…".

**Q20.1** Line 320: "save the Cubic": typo?

**R20.1** This line was removed after significant revision required using Cramer-von Mises test.

**Q21.1** Uncertainties are not quantified in your study which limits the interpretation of the implications. I am not asking to add this but at least mention this point somewhere in the discussion.

**R21.1** The line "The impact of sampling and distribution choice on uni- and multivariate return period values are presented, however uncertainties deserve further characterization." was added on Line 369.

**Q22.1** Similarly, it would be interesting to mention the limitation of this method in locations with multiple pluvial or coastal flood seasons.

**R22.1** The reviewer's suggestion has been adopted. Line 370 to 372 was added to address the limitations associated with multiple flood seasons. "This work focused solely on exploring conditional and joint probabilities of OWL and precipitation in a tidally and wave dominated semi-arid region and would not be applicable to regions experiencing multiple flooding seasons (e.g., Couasnon et al., 2022)".

**References**

- Couasnon, A., Sebastian, A., and Morales-Nápoles, O.: A copula-based Bayesian network for modeling compound flood hazard from riverine and coastal interactions at the catchment scale: An application to the Houston Ship Channel, Texas, Water, 10, 1190, 2018.
- Couasnon, A., Scussolini, P., Tran, T., Eilander, D., Muis, S., Wang, H., Keesom, J., Dullaart, J., Xuan, Y., Nguyen, H., et al.: A flood risk framework capturing the seasonality of and dependence between rainfall and sea levels–an application to Ho Chi Minh City, Vietnam, Water Resources Research, p. e2021WR030002, 2022.
- FEMA: Guidance for Flood Risk Analysis and Mapping: Coastal Flood Frequency and Extreme Value Analysis, https://www.fema.gov/sites/default/files/2020-02/Coastal_Flood_Frequency_and_Extreme_Value_Analysis_Guidance_Nov_2016.pdf, 2016.
- Genest, C., Rémillard, B., and Beaudoin, D.: Goodness-of-fit tests for copulas: A review and a power study, Insurance: Mathematics and economics, 44, 199–213, 2009.
- Moftakhari, H., Schubert, J. E., AghaKouchak, A., Matthew, R. A., and Sanders, B. F.: Linking statistical and hydrodynamic modeling for compound flood hazard assessment in tidal channels and estuaries, Advances in Water Resources, 128, 28–38, 2019.
- Sadegh, M., Moftakhari, H., Gupta, H. V., Ragno, E., Mazdiyasni, O., Sanders, B., Matthew, R., and AghaKouchak, A.: Multihazard scenarios for analysis of compound extreme events, Geophysical Research Letters, 45, 5470–5480, 2018.
- Santos, V. M., Wahl, T., Jane, R., Misra, S. K., and White, K. D.: Assessing compound flooding potential with multivariate statistical models in a complex estuarine system under data constraints, Journal of Flood Risk Management, 14, e12 749, 2021.
- Ward, P. J., Couasnon, A., Eilander, D., Haigh, I. D., Hendry, A., Muis, S., Veldkamp, T. I., Winsemius, H. C., and Wahl, T.: Dependence between high sea-level and high river discharge increases flood hazard in global deltas and estuaries, Environmental Research Letters, 13, 084 012, 2018.